# Boosting Text Encoder for Personalized Text-to-Image Generation

**NaHyeon Park**[*]                                                      *julia19@kaist.ac.kr*
*KAIST*

**Kunhee Kim**[*]                                                     *kunhee.kim@kaist.ac.kr*
*KAIST*

**Hyunjung Shim**                                                     *kateshim@kaist.ac.kr*
*KAIST*

**Reviewed on OpenReview:** *https://openreview.net/forum?id=hiZzk1nHuV*

## Abstract

In this paper, we introduce TEXTBOOST, an efficient one-shot personalization approach for text-to-image diffusion models. Traditional personalization methods typically involve fine-tuning extensive portions of the model, leading to substantial storage requirements and slow convergence. In contrast, we propose selectively fine-tuning only the text encoder, significantly improving computational and storage efficiency. To preserve the original semantic integrity, we develop a novel causality-preserving adaptation mechanism. Additionally, lightweight adapters are employed to locally refine text embeddings immediately before their interaction with cross-attention layers, greatly enhancing the expressiveness of text embeddings with minimal computational overhead. Empirical evaluations across diverse concepts demonstrate that TEXTBOOST achieves faster convergence and substantially reduces storage demands by minimizing the number of trainable parameters. Furthermore, TEXTBOOST maintains comparable subject fidelity, superior text fidelity, and greater generation diversity compared to existing methods. We show that our proposed method offers an efficient, scalable, and practically applicable solution for high-quality text-to-image personalization, particularly beneficial in resource-constrained environments.

## 1 Introduction

Recent advancements in text-to-image (T2I) diffusion models have significantly expanded the creative potential of AI-driven image synthesis, enabling high-fidelity images to be generated from natural language prompts (Ramesh et al., 2021; Rombach et al., 2022; Balaji et al., 2022; Nichol et al., 2023; Chen et al., 2024c; Podell et al., 2024; Esser et al., 2024). Despite these advances, standard text prompts often lack the specificity required to capture fine details or unique artistic styles, leading to ambiguities in the generated images. To address this, personalization techniques have emerged as a critical research direction, aiming to capture a specific concept ($V^\star$) within the token space so that the generated images faithfully reflect user-defined content (Gal et al., 2023; Ruiz et al., 2023).

While existing personalization methods show promising results, they often suffer from computational and storage inefficiencies. For instance, DreamBooth (Ruiz et al., 2023) fine-tunes all parameters of the U-Net and saves a distinct model for each concept, which demands substantial storage overhead. Moreover, current

---

*Equal contribution.

methods often require multiple reference images (usually three to five), limiting their practicality in real-world scenarios where users can only provide a single reference image. These issues underscore the need for efficient approaches to personalization—both in terms of parameter usage and training data requirements.

In this paper, we propose an efficient one-shot personalization method by focusing on an underexplored yet pivotal component of the T2I model: the text encoder. Typically, personalization approaches fix the text encoder and fine-tune components within the U-Net, such as cross-attention modules (Kumari et al., 2023; Chen et al., 2024b). However, our initial empirical analysis suggests that when all parameters of the T2I model are fine-tuned, the text encoder undergoes the largest changes, indicating significant potential benefits if adapted properly.

Building on this insight, we introduce **TextBoost**, a novel method that selectively fine-tunes the CLIP text encoder to achieve efficient one-shot personalization. A key challenge is to fine-tune the text encoder without disrupting its learned semantic space, particularly its auto-regressive causality: the property where a token's output embedding is conditioned only on itself and prior tokens in the sequence. To address this, we introduce Causality-Preserved Adaptation (CPA), a mechanism specifically designed to ensure that during the learning of a new concept token ($V^\star$), the output embeddings of all tokens preceding $V^\star$ remain unchanged. This preserves the original contextual understanding of the sequence leading up to the new concept. Concretely, CPA formulates the adapted output as a mixture of original and fine-tuned embeddings, preventing the distortion of established semantic structures of the original text encoder.

To further enhance fine-grained concept control, we draw inspiration from hierarchical representations in GAN architectures (Karras et al., 2019; 2020; Richardson et al., 2021; Tov et al., 2021) and introduce lightweight adapters that locally refine text embeddings before they interact with each cross-attention layer. Unlike token-space methods that rely on multiple forward passes (Voynov et al., 2023; Alaluf et al., 2023), our design requires only a single pass through these adapters, substantially improving computational efficiency and runtime.

We validate TEXTBOOST across various datasets, comparing it to popular personalization methods such as DreamBooth (Ruiz et al., 2023), Textual Inversion (Gal et al., 2023), LoRA (Hu et al., 2022), and Custom Diffusion (Kumari et al., 2023). Our results demonstrate that TEXTBOOST achieves faster convergence, reduced storage overhead, high subject and text fidelity, thus offering an improved balance between efficiency and generative quality.

In summary, our key contributions are as follows:

- **Fine-tuning text encoders for T2I personalization**: We present a novel approach for fine-tuning text encoders specifically designed for text-to-image (T2I) personalization, a previously unexplored direction.

- **Causality-preserved adaptation (CPA)**: We introduce a causality-preserved adaptation strategy that maintains the integrity of the text encoder's original semantic properties during adaptation, thereby preserving the causal relationships embedded within the semantic space.

- **Extended text embedding**: We propose extending the embedding space using lightweight adapters, facilitating more precise and fine-grained adaptation of concepts through enhancements in the cross-attention mechanism.

- **Efficient personalization**: Our method significantly improves training efficiency and reduces storage requirements, enabling high-quality T2I personalization suitable for deployment in resource-constrained real-world applications.

## 2 Related Work

### 2.1 Personalized text-to-image generation

Recent advancements in text-to-image (T2I) generation have increasingly emphasized personalization, allowing models to capture user-specific concepts more accurately. Textual Inversion (Gal et al., 2023) introduced

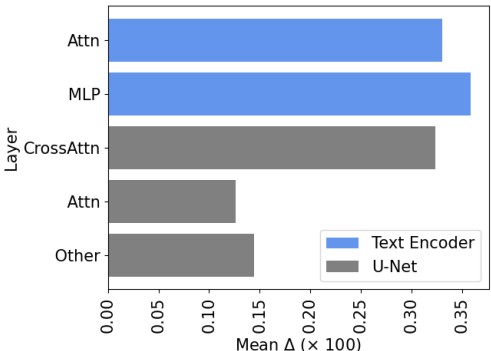

Figure 1: **Change in weights of different layers during fine-tuning of a T2I model.** The mean weight change in the text encoder layers is notably larger compared to that of the U-Net parameters. This suggests that the text encoder plays a pivotal role in the personalization task.

encoding personalized information into special learned tokens ($V^\star$ tokens). Building upon this, (Voynov et al., 2023) proposed a more expressive representation with layer-wise prompt embeddings for detailed concept inversion across different model layers. (Alaluf et al., 2023) further extended this paradigm with a timestep-wise inversion approach, offering greater flexibility but at the cost of increased computational overhead and longer training and inference times. Parallel efforts using fine-tuning-based methods, such as Dream-Booth (Ruiz et al., 2023), achieve superior fidelity by updating significant portions of the diffusion model, yet incur substantial computational and storage demands. To address these limitations, techniques like Custom Diffusion (Kumari et al., 2023) and OFT (Qiu et al., 2023) explored partial model fine-tuning, reducing computational load but still requiring modifications to the image generation component. Encoder-focused adaptations, including ELITE (Wei et al., 2023) and BLIP-Diffusion (Li et al., 2023), present another pathway to reduce computational overhead by primarily updating encoder components, thus minimizing trainable parameters while preserving personalized output quality. However, these encoder-based methods typically depend on extensive pre-training on large-scale datasets.

## 2.2 Parameter-efficient fine-tuning

Parameter-efficient fine-tuning (PEFT) methods have recently emerged as crucial techniques for effectively training large foundation models with limited resources. Adapter-based methods (Houlsby et al., 2019) insert lightweight layers into pretrained models, enabling efficient adaptation. Hu et al. (2022) introduced a simpler and highly efficient technique by inserting low-rank adaptation (LoRA) layers in parallel to existing weights, allowing the modifications to be merged back into the original model parameters after training. Due to its efficiency and effectiveness, LoRA has become popular in personalized T2I diffusion tasks, inspired subsequent studies (Qiu et al., 2023; Ruiz et al., 2024; Shah et al., 2024) and the basic component of recent personalization methods (Chen et al., 2024b; Lee et al., 2024). Similarly, StyleDrop utilized adapters for transformer-based diffusion models, demonstrating effective customization capabilities (Sohn et al., 2023). Our proposed method also leverages techniques from the PEFT domain, such as adapters (Houlsby et al., 2019) and LoRA (Hu et al., 2022). However, unlike previous approaches primarily aimed at optimizing downstream task performance, our method emphasizes preserving the original capabilities of pretrained models by adaptively fine-tuning parameters in a causality-preserved manner.

## 3 Motivation

Unlike previous works, which typically fine-tune either the concept token $V^\star$ or specific components of the U-Net, we begin by systematically re-evaluating which parts of the text-to-image (T2I) diffusion model are most critical to fine-tuning for effective personalization.

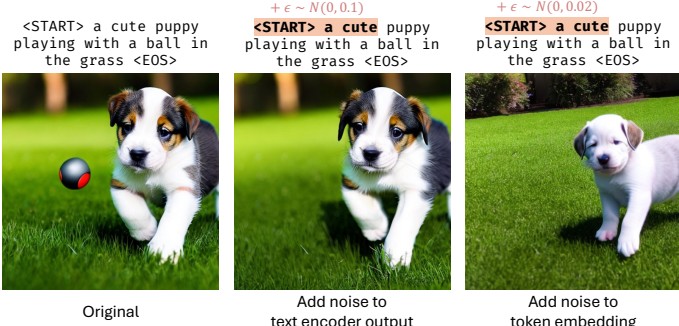

Figure 2: **Effect of perturbations on early token embeddings during text-to-image generation.** We compare the original generation (left) with images produced after adding small Gaussian noise to the text encoder's output embeddings (middle) and directly to the token embeddings (right). Adding even slight noise significantly affects generated image quality, highlighting the sensitivity and importance of preserving embeddings for early tokens in text-to-image generation.

Kumari et al. (2023) proposed an efficient approach by fine-tuning only the cross-attention layers of the U-Net, motivated by observing substantial parameter changes within these layers during personalization (Li et al., 2020). Specifically, they targeted the key and value projection layers. However, their analysis did not consider potential changes in the text encoder parameters, a critical component responsible for generating embeddings fed into the cross-attention layers.

Given that key and value projections directly depend on text embeddings, we hypothesized that the text encoder might also significantly influence personalization outcomes. To rigorously test this hypothesis, we extended the investigation from (Kumari et al., 2023) to include both the U-Net and the text encoder. We quantified parameter changes during fine-tuning using the following measure:

$$\Delta = \frac{||\hat{\theta} - \theta||}{||\theta||}, \tag{1}$$

where $\theta$ and $\hat{\theta}$ denote the original and fine-tuned weights.

Consistent with previous findings (Kumari et al., 2023), we observed significant changes in the U-Net's cross-attention layer parameters. Surprisingly, as illustrated in Fig. 1, the text encoder parameters exhibited even more substantial adjustments. This finding strongly highlights the previously overlooked importance of the text encoder in personalized T2I generation, aligning with recent insights emphasizing its central role (Saharia et al., 2022; **?**; Li et al., 2024b).

To the best of our knowledge, this study is the first systematic exploration specifically targeting fine-tuning the text encoder for personalized text-to-image generation.

## 4 Boosting text encoder tuning

Building upon insights from the preceding section, we introduce TEXTBOOST, a novel approach for personalizing text-to-image diffusion models.

### 4.1 Preliminaries: Text-to-image diffusion models

Diffusion models (Sohl-Dickstein et al., 2015) have recently become the most widely adopted generative models for text-to-image generation (Saharia et al., 2022; Ramesh et al., 2022; Rombach et al., 2022). These models aim to closely approximate the original data distribution $q(\boldsymbol{x}_0)$ with $p_\theta(\boldsymbol{x}_0)$. Here, $p_\theta(\boldsymbol{x}_0) \coloneqq \int p_\theta(\boldsymbol{x}_{0:T})$, where $p_\theta(\boldsymbol{x}_{0:T})$ is termed as the reverse process being Markov chain with learned Gaussian transitions. The approximate posterior $q(\boldsymbol{x}_{1:T}|\boldsymbol{x}_0)$ is termed a forward process, of which the noise is gradually added to the original data point $\boldsymbol{x}_0$ as $\boldsymbol{x}_t = \sqrt{\alpha_t}\boldsymbol{x}_0 + \sqrt{1-\alpha_t}\epsilon$, can be expressed as $q(\boldsymbol{x}_{1:T}|\boldsymbol{x}_0) \coloneqq \prod_{t=1}^{T} q(\boldsymbol{x}_t|\boldsymbol{x}_{t-1})$. The

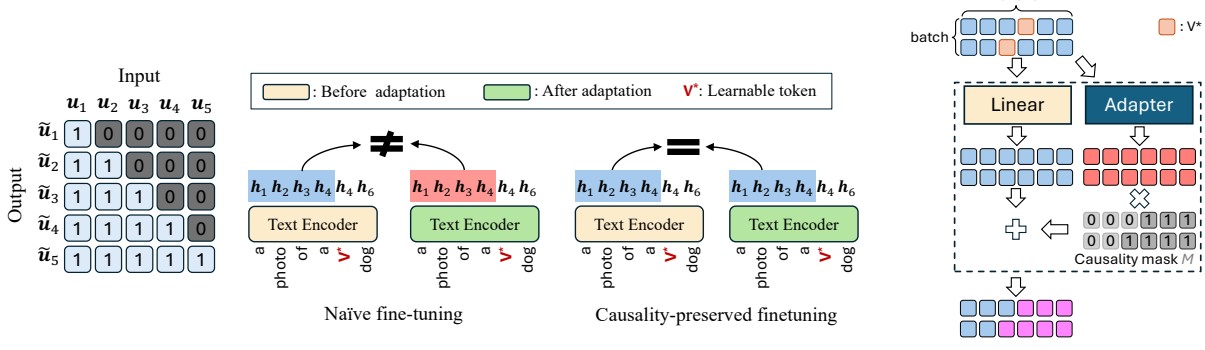

(a) Causal attention mask

(b) Causality of CLIP text encoder

(c) Causality-preserved adapter

Figure 3: **(a) Causal attention mask.** The CLIP text encoder employs causal masking techniques, where the output embedding $\boldsymbol{h}_i$ corresponding to the input token $\boldsymbol{y}_i$ can only attend to preceding tokens. **(b) Causality of CLIP text encoder.** If this causality is not properly accounted for during fine-tuning (e.g., through naive fine-tuning), the embeddings of tokens preceding the new concept token $\mathtt{V}^\star$ can be unintentionally altered, leading to undesired changes in their representation. **(c) Our causality-preserved adapter.** We attach adapters in parallel to the linear layers, and incorporate a causality mask to ensure that the fine-tuned adapter influences only the new concept token $\mathtt{V}^\star$ and the subsequent tokens, thereby maintaining the causality structure.

training objective is formulated with variational bound on negative log-likelihood:

$$\mathbb{E}[-\log p_\theta(\boldsymbol{x}_0)] \leq \mathbb{E}_q[-\log \frac{p_\theta(\boldsymbol{x}_{0:T})}{q(\boldsymbol{x}_{1:T}|\boldsymbol{x}_0)}] := \mathcal{L}, \tag{2}$$

which is further simplified into the following objective:

$$\mathcal{L}_{simple}(\theta) := \mathbb{E}_{t,\boldsymbol{x}_0,\epsilon}[||\epsilon - \epsilon_\theta(\boldsymbol{x}_t,t)||_2^2]. \tag{3}$$

Text-to-image (T2I) diffusion models (Rombach et al., 2022) take text condition $\mathbf{c}$ as additional input, where the text prompt $y$ is encoded with text encoder $\mathcal{E}_T$ as $\mathbf{c} = \mathcal{E}_T(y)$. These models usually operate in latent space, encoding an image $\boldsymbol{x}_0$ into a latent $\boldsymbol{z}_0 = \mathcal{E}_I(\boldsymbol{x}_0)$. The forward diffusion process is then applied in latent space, yielding $\boldsymbol{z}_t = \sqrt{\alpha_t}\boldsymbol{z}_0 + \sqrt{1-\alpha_t}\epsilon$. The training objective hence takes an additional $\mathbf{c}$ and minimizes the following loss:

$$\mathbb{E}_{\boldsymbol{z}_0,t,y,\epsilon}\left[||\epsilon - \epsilon_\theta(\boldsymbol{z}_t,t,\mathbf{c})||_2^2\right]. \tag{4}$$

While our approach generalizes to various T2I models, we primarily employ Stable Diffusion (Rombach et al., 2022), due to its extensive usage and open accessibility. Stable Diffusion adopts a U-Net architecture (Ronneberger et al., 2015) for its denoising diffusion process. To guide the image generation, text embeddings are injected through cross-attention layers, which dynamically integrate these embeddings into the U-Net at each diffusion step.

Given the critical role of the CLIP text encoder (Radford et al., 2021) in widely-used T2I diffusion models such as Stable Diffusion (SD) (Rombach et al., 2022) and Stable Diffusion XL (SDXL) (Podell et al., 2024), we specifically targeted the CLIP text encoder for effective personalization in our method.

## 4.2 Causality-preserved adaptation

### 4.2.1 Challenges in fine-tuning CLIP text encoder

The CLIP text encoder, like many language models, is a decoder-only transformer (Radford et al., 2021) that employs causal masking (see Fig. 3(a)). This architectural feature enforces an auto-regressive processing

flow: the output embedding $\boldsymbol{h}_i$ for an input token $\boldsymbol{y}_i$ is generated based solely on $\boldsymbol{y}_i$ and all preceding tokens $(\boldsymbol{y}_1, \ldots, \boldsymbol{y}_{i-1})$. It is not influenced by any subsequent tokens ($\boldsymbol{y}_j$ where $j > i$). This inherent sequential causality is fundamental to how the encoder builds contextual representations.

When fine-tuning the encoder to learn a new concept token $\mathtt{V}^\star$, a naive update to the model's weights can inadvertently alter the output embeddings of tokens preceding $\mathtt{V}^\star$ (as illustrated in Fig. 3(b) where $h_1$ to $h_4$ change). This happens because weight changes, even if optimized for $\mathtt{V}^\star$, propagate throughout the model. Such alterations disrupt the established semantic interpretations of the initial part of the text sequence (see Fig. 2), which can degrade the model's ability to understand prompts and accurately depict the concept. Our goal is to adapt the model for $\mathtt{V}^\star$ while strictly preserving the original output embeddings of all tokens $\boldsymbol{h}_i$ that appear before $\mathtt{V}^\star$ in any given prompt.

### 4.2.2 Causality-preserved adapter

One straightforward solution would involve fully fine-tuning the text encoder and subsequently overwriting embeddings preceding $\mathtt{V}^\star$ with their original values. This approach, however, requires two full forward passes, leading to inefficiencies and a failure to maintain layer-wise causality. Instead, we propose a more efficient causality-preserving adapter (CPA) inspired by parameter-efficient fine-tuning methods (Houlsby et al., 2019; Hu et al., 2022) from large language models (LLMs). We attach adapters in parallel to selected linear layers within the Transformer. Let $\boldsymbol{U} = [\boldsymbol{u}_1; \ldots; \boldsymbol{u}_n] \in \mathbb{R}^{n \times d_{\text{in}}}$ denote the matrix of token representations entering an adapted linear layer. The adapted output is computed as

$$\widetilde{\boldsymbol{U}} = \boldsymbol{U}\boldsymbol{W} + \text{Diag}(\boldsymbol{m})\mathcal{A}_{\text{CPA}}(\boldsymbol{U}), \tag{5}$$

where $\boldsymbol{W} \in \mathbb{R}^{d_{\text{in}} \times d_{\text{out}}}$ is the frozen pretrained weight matrix and $\mathcal{A}_{\text{CPA}}(\boldsymbol{U}) = \boldsymbol{U}\boldsymbol{W}_{down}\boldsymbol{W}_{up}$ represents the adapter with LoRA-style down-projection $\boldsymbol{W}_{down} \in \mathbb{R}^{d_{in} \times r}$ and up-projection $\boldsymbol{W}_{up} \in \mathbb{R}^{r \times d_{out}}$. For a tokenized prompt $\boldsymbol{Y} = [\boldsymbol{y}_1; \ldots; \boldsymbol{y}_n]$, with the special token $\mathtt{V}^\star$ located at position $i^\star$, the causal mask vector $\boldsymbol{m} = [m_1, \ldots, m_n]^\top \in \{0, 1\}^n$ is defined as

$$m_i = \begin{cases} 1, & \text{if } i \geq i^\star \text{ and } \boldsymbol{y}_i \text{ is not a padding token,} \\ 0, & \text{otherwise.} \end{cases} \tag{6}$$

For a batch of $B$ prompts, these vectors can be stacked row-wise into a batched mask matrix $\boldsymbol{M}_{\text{CPA}} \in \{0, 1\}^{B \times n}$, which is the form illustrated in Fig. 3(c). This mask ensures that the adapter affects only $\mathtt{V}^\star$ and the subsequent non-padding tokens. For all tokens with $i < i^\star$, we have $m_i = 0$, so the $i$-th row of the adapter term in Eq. 5 is zero and the adapted output reduces to the pretrained output. This preserves the original encoder representations for the initial part of the sequence and maintains the auto-regressive causality up to $\mathtt{V}^\star$ (see Fig. 3(c)). We call this overall strategy Causality-Preserved Adaptation (CPA).

Based on empirical findings and our ablation studies (see Fig. 10), we incorporated adapters into specific layers for optimal performance. Notably, our approach is different from some previous conventions that primarily target self-attention parameters with adapters[1]. Instead, our ablations indicated that focusing on other components yielded better results for our personalization task. Specifically, within Multi-Layer Perceptron (MLP) blocks of the text encoder, we add adapters to the second feed-forward layer (`fc2`). The rank parameter $r$ determines the adapter's bottleneck dimension, controlling both parameter efficiency and expressivity; unless otherwise stated, we adopt $r = 1$ as our default configuration.

## 4.3 Extended textual embedding

Extended latent representations have proven effective in achieving more precise inversion in generative adversarial networks (Goodfellow et al., 2014; Karras et al., 2019), as demonstrated by methods like Image2StyleGAN (Abdal et al., 2019) or e4e (Tov et al., 2021). Motivated by these advances, we propose to extend the CLIP text encoder into an extended representation space.

---

[1] https://github.com/huggingface/diffusers/blob/main/examples/dreambooth/train_dreambooth_lora.py

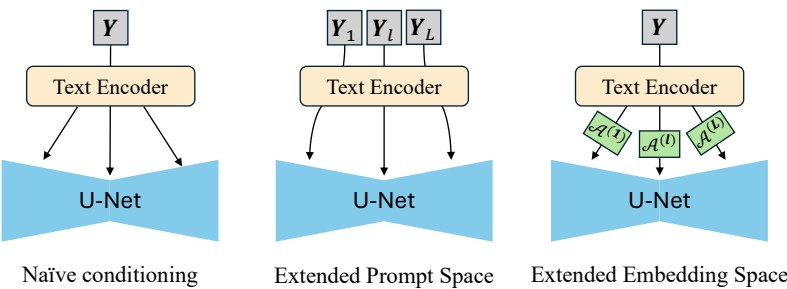

Figure 4: **Extended embedding space.** Unlike previous methods that operate in the prompt space, our approach introduces adapters in the embedding space, resulting in an extended embedding space. This design allows for more flexible integration of the fine-tuned text embedding from the text encoder into the image generation module. Here, $\boldsymbol{Y}$ represents the input text prompt, and $\mathcal{A}^{(\ell)}$ denotes the adapter for cross-attention layer $\ell$.

Let a tokenized prompt $\boldsymbol{Y} = [\boldsymbol{y}_1; \dots; \boldsymbol{y}_n]$ be encoded by the CLIP text encoder $\mathcal{E}_T$, producing final hidden states

$$\boldsymbol{H} = \mathcal{E}_T(\boldsymbol{Y}) \in \mathbb{R}^{n \times d}. \tag{7}$$

In standard text-to-image diffusion models, this single embedding matrix $\boldsymbol{H}$ is shared across all cross-attention layers of the U-Net, where it is projected to keys and values.

We define *extended textual embedding* as a layer-wise conditioning scheme that replaces the single text-encoder hidden state $H$ with a set of per-cross-attention embeddings $\{\boldsymbol{H}^{(\ell)}\}_{\ell=1}^{L_{\mathrm{ca}}}$ where $L_{\mathrm{ca}}$ denotes the number of cross-attention layers in the U-Net.

$$\boldsymbol{H}^{(\ell)} = \boldsymbol{H} + \mathrm{Diag}(\boldsymbol{m})\mathcal{A}_\ell(\boldsymbol{H}), \tag{8}$$

where $\mathcal{A}_\ell(\cdot)$ denotes a lightweight adapter specific to cross-attention layer $\ell$, and $\boldsymbol{m} \in \{0,1\}^n$ is the causality mask defined in Section 4.2.

Each $\boldsymbol{H}^{(\ell)}$ is injected into cross-attention block $\ell$ *before* its key and value projection layers:

$$\boldsymbol{K}_\ell = \boldsymbol{H}^{(\ell)} \boldsymbol{W}_\ell^K, \qquad \boldsymbol{V}_\ell = \boldsymbol{H}^{(\ell)} \boldsymbol{W}_\ell^V. \tag{9}$$

Importantly, the projection matrices are unchanged; instead, each cross-attention layer receives its own adapted textual embedding. This effectively extends the conditioning space from a single embedding $\boldsymbol{H}$ to a collection $(\boldsymbol{H}^{(1)}, \dots, \boldsymbol{H}^{(L_{\mathrm{ca}})})$, increasing representational flexibility while maintaining architectural simplicity.

Crucially, to preserve causality, embeddings corresponding to tokens positioned before the novel concept token $\mathtt{V}^\star$ remain unchanged due to the masking term $\mathrm{Diag}(\boldsymbol{m})$.

As illustrated in Fig. 4, we apply causality-preserved adaptation (CPA) to refine text embeddings separately for each cross-attention layer. Unlike token-space methods such as XTI (Voynov et al., 2023) or NeTI (Alaluf et al., 2023), which require multiple forward passes through the text encoder, our approach performs only one forward pass through the text encoder followed by lightweight adapter transformations to construct $\{\boldsymbol{H}^{(\ell)}\}_{\ell=1}^{L_{\mathrm{ca}}}$. This design significantly reduces computational overhead and improves efficiency compared to hierarchical prompt-embedding techniques.

We refer to the text-encoder fine-tuning approach with CPA as TEXTBOOST, and the extended version incorporating both CPA and extended textual embedding as TEXTBOOST++.

Table 1: **Quantitative comparison on Stable Diffusion v2.1-base.** We measure VQA scores (Lin et al., 2024) for image-text fidelity. Pairwise DINOv2 (Oquab et al., 2024) feature cosine similarity is evaluated with the reference images that are not used for training. For practicality, we compare the number of parameters.

| Methods | VQA ↑ | DINOv2 ↑ | Diversity ↑ | # Params ↓ |
|---|---|---|---|---|
| Textual Inversion (Gal et al., 2023) | 0.475 | 0.467 | **0.408** | 0.00M |
| NeTI (Alaluf et al., 2023) | 0.602 | 0.536 | 0.279 | 0.83M |
| DreamBooth (Ruiz et al., 2023) | 0.553 | 0.587 | 0.164 | 865.91M |
| DreamBooth-LoRA (Hu et al., 2022) | 0.528 | 0.589 | 0.190 | 0.83M |
| Custom Diffusion (Kumari et al., 2023) | 0.452 | **0.632** | 0.136 | 25.56M |
| DisenBooth (Chen et al., 2024b) | 0.675 | 0.486 | 0.305 | 2.93M |
| TEXTBOOST | **0.680** | 0.560 | 0.307 | **0.12M** |
| TEXTBOOST++ | 0.607 | 0.598 | 0.229 | **0.15M** |

Table 2: **Quantitative comparison with encoder-based methods.** We compare TEXTBOOST with encoder-based methods, which require pre-training. The results highlight the superior performance of our method over the baselines in terms of subject similarity and text fidelity.

| Methods | T2I Model | VQA ↑ | DINOv2 ↑ |
|---|---|---|---|
| ELITE | SD1.4 | 0.430 | 0.489 |
| TEXTBOOST++ | | **0.558** | **0.564** |
| BLIP Diffusion | SD1.5 | **0.650** | 0.531 |
| TEXTBOOST++ | | 0.577 | **0.575** |

# 5 Experiments

## 5.1 Experimental setup

**Dataset.** We employed the benchmark introduced by Ruiz et al. (2023), which consists of 30 subjects and 25 text prompts, with each subject having 4 to 6 associated images. For style experiments, we use the reference images provided by StyleDrop (Sohn et al., 2023). It is important to note that we trained all of the models using a single reference image. Following DreamBooth, we generated 4 images per prompt (3,000 images in total) for evaluation.

**Evaluation metrics.** We evaluate image-text alignment, which measures how well the image reflects the user prompt, and subject fidelity, which assesses how accurately the image represents the characteristics of the given concept. Regarding image-text alignment, while prior works commonly used CLIP score, recent findings by (Lin et al., 2024) indicate that the VQA score better reflects human perception. Therefore, we opted for the VQA score for a more accurate evaluation. Subject fidelity was assessed by computing the pairwise cosine similarity between generated and reference image embeddings, following the methodology of (Ruiz et al., 2023). We employed `DINOv2:ViT/L-14` from DINOv2 (Oquab et al., 2024) as our feature extractor, aligning with recent approaches (Lee et al., 2024; Jang et al., 2024). We used the foreground of the validation images as suggested by (Kim et al., 2024). To obtain foreground images, we employed GroundingDINO (Liu et al., 2024) and GroundedSAM (Ren et al., 2024). Since these scores do not evaluate the extent of overfitting, memorization of reference images can result in inflated scores. Considering this, we assessed the inter-similarity between generated images using the DINOv2 (Oquab et al., 2024) embedding. Finally, to assess the practical applicability of our method, we conducted a user study involving 50 participants on Amazon Mechanical Turk to evaluate user preferences.

**Baselines.** We compare our approach with recent personalization methods: DreamBooth (Ruiz et al., 2023), DreamBooth-LoRA (Hu et al., 2022), Textual Inversion (Gal et al., 2023), Custom Diffusion (Kumari et al., 2023), NeTI (Alaluf et al., 2023) and DisenBooth (Chen et al., 2024b). We also compare our

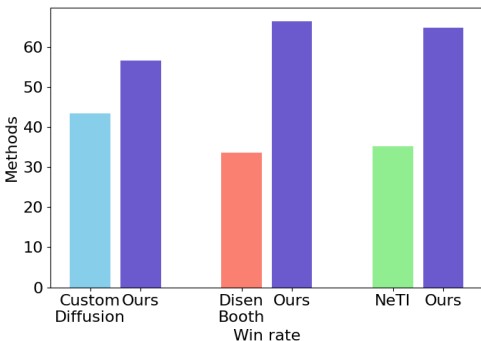

Figure 5: **User study.** We conducted a large-scale user-preference survey with 50 participants on Amazon Mechanical Turk. We asked each participant 30 questions to choose an image that best resembles the given subject and aligns with the text prompt at the same time (1,500 responses in total).

method with zero-shot personalization methods (Wei et al., 2023; Li et al., 2023), which require pre-training beforehand.

**Implementation details.** We utilized the `imagenet small` template for the training prompt, following (Gal et al., 2023). For the identifier, we employed templates of the form `V* [class]`, similar to Custom Diffusion. For example, our prompt was constructed as, "a photo of a nice `V* dog`". We also implemented weak augmentation, though significantly less aggressive than that used in Custom Diffusion (Kumari et al., 2023). Furthermore, we did not modify the loss computation region using masking. For the T2I model, we used various versions of Stable Diffusion (Rombach et al., 2022) model, integrating the text encoder from the CLIP (Radford et al., 2021) or OpenCLIP (Cherti et al., 2023). Our adapter is trained using the AdamW optimizer (Loshchilov & Hutter, 2019) with a learning rate of 1e-4 for text encoder and 1e-3 for $V^*$ token for 100 steps. For TEXTBOOST++, we trained for 80 steps. All experiments were conducted with batch size 8 on a single NVIDIA A6000 GPU.

## 5.2 Quantitative results

The quantitative comparison is presented in Tab. 1. Our method performs on par with existing approaches while using significantly fewer parameters. TEXTBOOST achieves a notably high VQA score, demonstrating strong performance. Regarding subject fidelity, our method consistently produces results that are comparable to or exceed those of other methods, suggesting that with proper tuning of the text encoder, effective personalization can be achieved without sacrificing performance. Additionally, our method excels in generating a diverse range of output images, as indicated by the high diversity score, providing end users with diverse options.

We also compare our method, TEXTBOOST, with ELITE and BLIP Diffusion in Tab. 2. As shown in the table, TEXTBOOST achieves comparable or even superior image and text fidelity compared to these encoder-based methods, which typically require extensive pre-training. This highlights a key advantage of our approach, as it does not require pre-training or subject data, enabling rapid and efficient fine-tuning during test time.

**Computational efficiency.** We emphasize the practical advantages of our method, which demands significantly fewer trainable parameters, as detailed in Tab. 1. In this regard, the fact that our method achieves performance comparable to existing approaches is a significant advantage. Moreover, personalization introduces storage challenges, as the fine-tuned model must be saved for each concept. Consequently, the number of parameters directly correlates with storage requirements, as all parameters related to the fine-tuned components of the T2I model must be stored. While Textual Inversion offers notable storage efficiency due to its focus on fine-tuning text embeddings, its limited performance hinders practical application, as evidenced by the low image fidelity and image-text alignment scores shown in the table. The compact size of our model

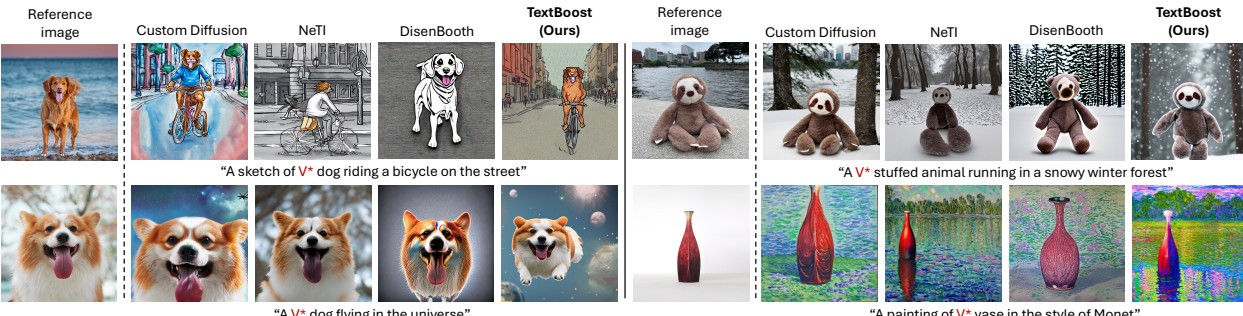

Figure 6: **Qualitative comparison.** We compare images generated by a diverse set of methods, utilizing various types of text prompts across different subjects. It is important to note that all models were trained using a *single* reference image, positioned at the leftmost of each comparison. The results demonstrate that our TEXTBOOST method can accurately generate images that adhere to the given prompt, maintaining high subject fidelity, even when compared to more computationally intensive methods with a large number of parameters.

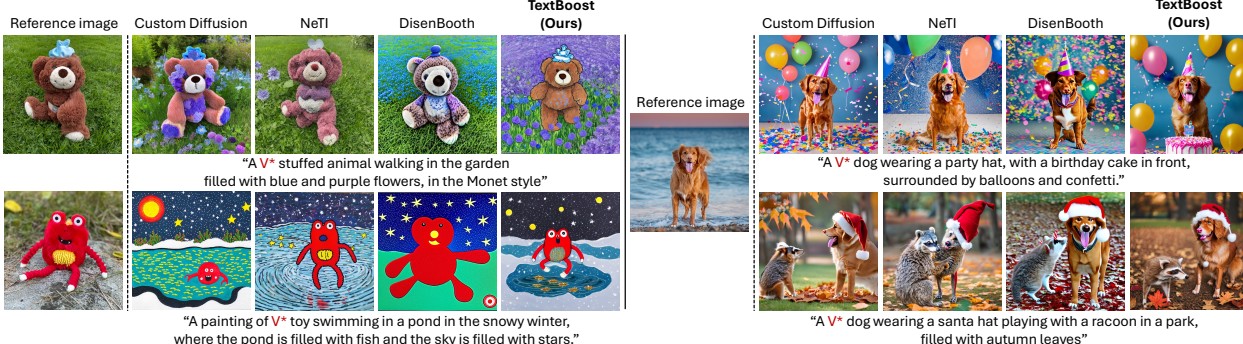

Figure 7: **Comprehensive captions.** To demonstrate the effectiveness of TEXTBOOST in handling detailed captions, we generate images from long and creative descriptions that vary in attributes, backgrounds, and styles. Compared to other baseline methods, TEXTBOOST exhibits superior performance in accurately reflecting these comprehensive captions, maintaining high subject fidelity.

allows it to be seamlessly stored in environments with limited storage capacity, such as the cloud or portable storage devices.

**User study.** In real-world scenarios, users need to select the output image that both satisfies (1) subject fidelity and (2) alignment between the text prompt and the generated image. To evaluate existing methods and understand user preferences, we conducted a large-scale user study. Participants were tasked with selecting the best image from multiple options generated by different methods. We employed diverse subjects and text prompts sourced from (Ruiz et al., 2023) (templates in Appendix A). To ensure fairness, random seeds were fixed, and we computed the win-rate between our method and previous works. Through Amazon Mechanical Turk, 50 participants completed 30 questions each, yielding a total of 1,500 responses. As indicated in Fig. 5, our method showed higher win-rates compared to all of the existing methods, demonstrating its superior ability to meet user demands for subject fidelity and text-image alignment in practical settings.

### 5.3 Qualitative results

We conducted a qualitative analysis to compare the baseline models with TEXTBOOST, and the results are presented in Fig. 6 and Fig. 7.

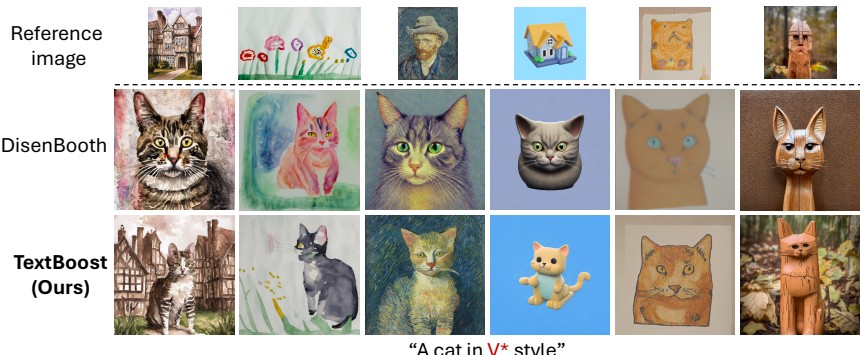

Figure 8: **Results on stylization.** The output generated by TEXTBOOST effectively captures the reference style while maintaining key details such as texture, color patterns, and artistic nuances. In comparison to the recent method, DisenBooth, our TEXTBOOST demonstrates superior fidelity in reproducing these stylistic features.

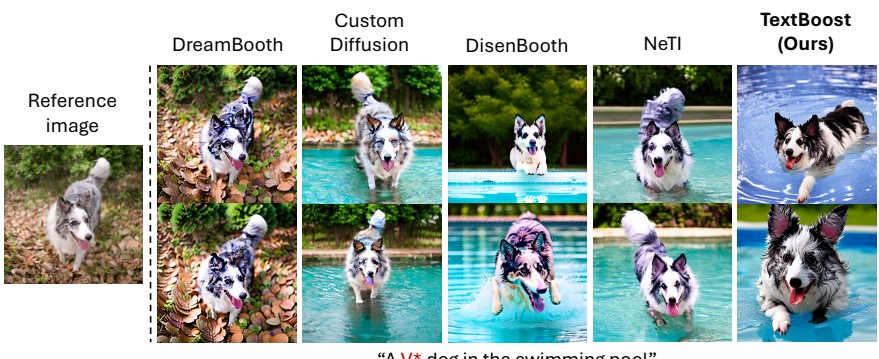

Figure 9: **Diversity comparison.** We evaluate the inter-similarity of images generated with identical prompts across various baselines. Our TEXTBOOST consistently generates images with high diversity.

To begin with, Fig. 6 illustrates examples generated from creative text prompts. TEXTBOOST consistently outperforms other methods in accurately reflecting the user's prompt. For instance, in the prompt "A V* dog riding a bicycle on the street", alternative models exhibit several issues, such as the omission of the bicycle (DisenBooth), the incorrect display of only the bicycle (NeTI), or the occurrence of attribute leakage (e.g., a blue shirt and a street in the image from Custom Diffusion). In contrast, TEXTBOOST correctly includes all the elements specified in the prompt. More qualitative comparison results can be found in Appendix B.

We further evaluated TEXTBOOST with more complex prompts that involve simultaneous modifications to attributes, background, and style, as depicted in Fig. 7. Our findings demonstrate that TEXTBOOST accurately interprets and generates images based on these comprehensive captions. For example, the term "walking" in the top-left prompt posed difficulties for baseline models in capturing the requisite details and style, whereas TEXTBOOST effectively represented these elements. In the bottom-right example, the inclusion of the word "raccoon" caused confusion in other models, resulting in artifacts or misidentification of a dog. However, TEXTBOOST accurately generated the output of both the dog and raccoon as intended.

These results indicate that TEXTBOOST enhances prompt alignment and ensures subject accuracy, making it an ideal tool for users seeking greater control over the creative process. Additional results, including experiments on facial datasets, can be found in Appendices C and E.

**Stylization.** We also conducted experiments on style personalization using a single reference image. Leveraging the style references from (Sohn et al., 2023) on Stable Diffusion v2.1, we evaluated our approach in

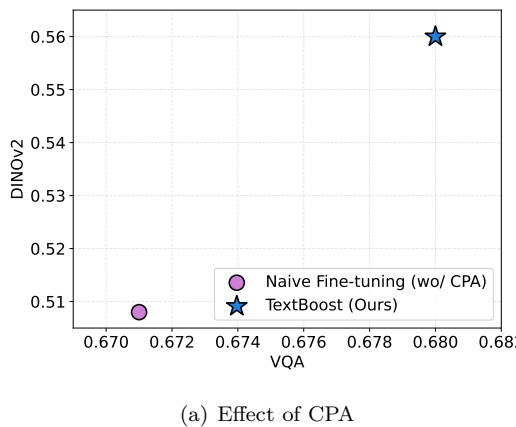

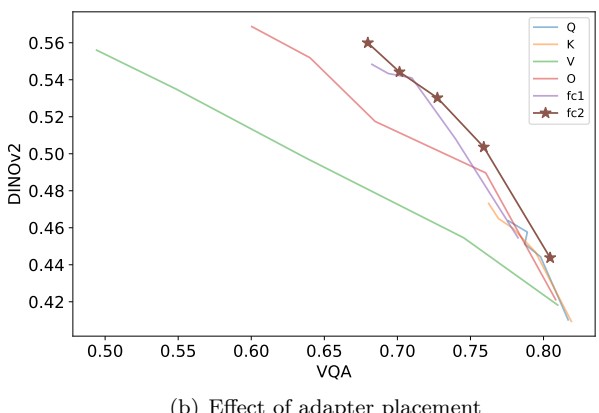

(a) Effect of CPA

(b) Effect of adapter placement

Figure 10: **Ablation study.** (a) Our proposed CPA results in better image and text fidelity (b) Placing the adapter in `fc2` shows best performance.

comparison to DisenBooth (Chen et al., 2024b), as alternative methods proved ineffective for this particular task. For example, NeTI tends to overfit, resulting in images that fail to capture the intended subject—such as a cat—while only reflecting the style (see Fig. 14 in Appendix B). As demonstrated in Fig. 8, our method effectively learns and applies the style from a single reference image, maintaining fidelity to the content of the text prompt.

**Diversity.** In Tab. 1, we measured the inter-similarity of output images using DINOv2, finding that our TEXTBOOST exhibits greater diversity than other approaches. A visual example is shown in Fig. 9, where other baselines produce similar images for the same prompts, while our method generates more varied poses and backgrounds. This demonstrates TEXTBOOST's advantage in generating diverse outputs, highlighting its strong potential for real-world applications. More results can be found in Appendix C.

### 5.4 Ablation study

Finally, we conduct an ablation study on our causality-preserved adaptation (CPA) and the adapter placement.

The ablation results show the effectiveness of our CPA; its removal from TEXTBOOST markedly lowered VQA and DINOv2 scores, demonstrating its significant contribution (see Fig. 10(a)). Note that Tab. 1 details our extended embedding space's effect.

While parameter-efficient fine-tuning (PEFT) often targets self-attention modules (e.g., LoRA), systematic studies of optimal PEFT targets are lacking. We thus explored six linear layers in transformer blocks: self-attention query (`Q`), key (`K`), value (`V`), output (`O`), and MLP layers (`fc1`, `fc2`). As shown in Fig. 10(b), MLP layer updates results in better text-image alignment compared to adapting self-attention such as `K` and `V` projections, verifying our integration of adapters into the MLP's `fc2` layer.

### 5.5 Results on SDXL

In widely used models such as Stable Diffusion v1.5 and v2.1, a single CLIP text encoder is used. However, the larger SDXL model has two text encoders: CLIP ViT-L and OpenCLIP ViT-bigG (Cherti et al., 2023). To test the scalability of our approach, we tested TEXTBOOST on larger models. Specifically, we fine-tune only the OpenCLIP ViT-bigG encoder of SDXL while keeping the CLIP ViT-L encoder frozen. This choice is motivated by the ablation results in Tab. 3, which suggest that adapting a single text encoder provides a favorable trade-off between personalization performance and computational efficiency. The results, shown in Fig. 11, demonstrate that our TEXTBOOST method effectively captures subject and style characteristics

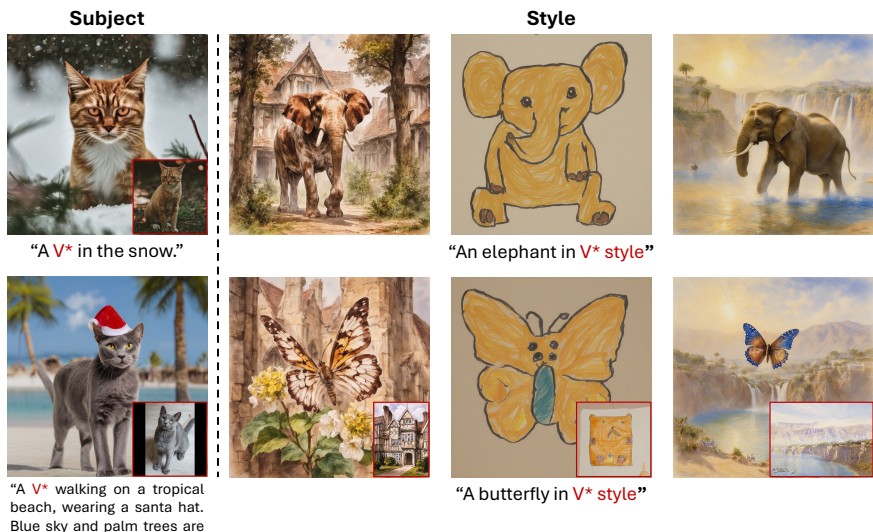

Figure 11: **Results on SDXL.** The results show that our proposed TEXTBOOST efficiently scales to larger T2I models, which require multiple text encoders while maintaining a high level of efficiency (a small number of trainable parameters and low storage requirements), which is practical, especially for large T2I models.

across a variety of prompts. In our GPU setup, methods like DreamBooth and Custom Diffusion require too many parameters, making them less efficient for personalization due to memory limits. On the other hand, our method uses far fewer parameters, enabling faster training and better T2I personalization within SDXL. This shows that our approach is more practical and better suited for resource-limited environments, such as real-world applications, research, or edge devices.

## 6 Discussion & Conclusion

The role of text encoders in large-scale text-to-image (T2I) models has attracted growing attention in recent years. Prior works (Saharia et al., 2022; Chen et al., 2024a; Li et al., 2024a) has shown that strengthening the text side of the model can yield substantial gains in generation quality and text-image alignment. Despite these advances, the potential of text encoder adaptation for personalization has remained largely underexplored, with most existing methods focusing instead on the image generation module or on learning concept-specific token representations.

In this work, we revisit this design choice and show that the text encoder is not merely an auxiliary conditioning component, but a highly effective locus for personalization. Through TEXTBOOST, we demonstrate that carefully adapting the text encoder can achieve personalization performance comparable to updating the image module, while maintaining strong diversity and offering a substantially more efficient parameterization. These findings suggest that high-quality one-shot personalization does not necessarily require extensive modification of the diffusion backbone; rather, significant gains can be achieved by selectively and structurally adapting the text-side representation space.

More broadly, our results highlight the importance of understanding how personalization is encoded and propagated through the conditioning pathway of T2I models. We believe this perspective opens up new opportunities for designing lightweight, scalable, and practically deployable personalization methods, particularly in resource-constrained settings where storage and training efficiency are critical. Looking forward, we expect that further investigation into text-side adaptation—including richer encoder architectures, more expressive conditioning schemes, and broader concept domains such as faces and artistic styles—will lead to more flexible and reliable personalization systems. We hope TEXTBOOST serves as a step toward this direction and stimulates further research on text-centric approaches to efficient T2I personalization.

## Version note

A preliminary arXiv version of this work (`arXiv:2409.08248v1`) used an earlier formulation based on augmentation tokens, knowledge preservation, and SNR-weighted timestep sampling. The present TMLR version substantially revises the method by introducing Causality-Preserved Adaptation (CPA) and the extended textual embedding variant TEXTBOOST++, together with updated experiments, ablations, and additional analyses. At the same time, both versions share the same central idea: fine-tuning the text encoder for efficient one-shot text-to-image personalization.

## Acknowledgements

This work was supported by the Institute for Information & communications Technology Planning & Evaluation (IITP) grant funded by the Korea government (MSIT) (RS-2019-II190075, Artificial Intelligence Graduate School Program (KAIST)); this research was supported by the Basic Science Research Program through the National Research Foundation of Korea (NRF) funded by the MSIP (No. RS-2025-00520207); by the IITP grant funded by the Korea government (MSIT) and KEIT grant funded by the Korea government (MOTIE) (No. 2022-0-01045); by the IITP grant funded by the Korea government (MSIT) and KEIT grant funded by the Korea government (MOTIE) (No. 2022-0-00680); and by the Institute of Information & communications Technology Planning & Evaluation (IITP) grant funded by the Korea government (MSIT) (No. RS-2024-00457882, National AI Research Lab Project).

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

# Appendix

## A    User Study Details

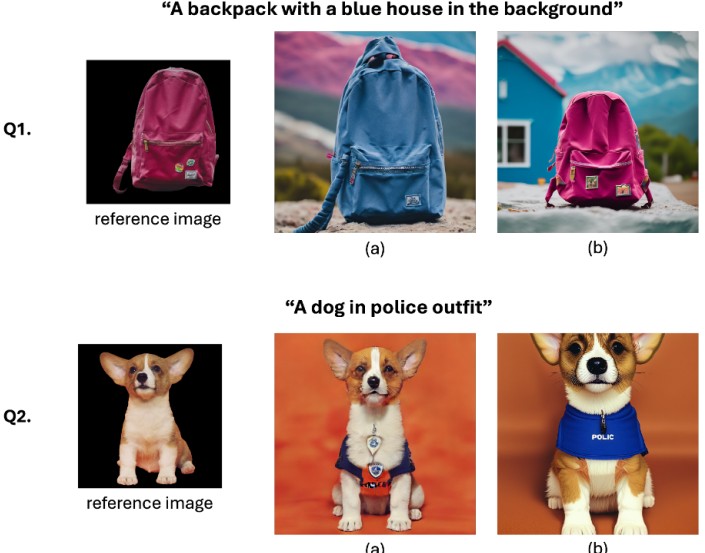

Figure 12: **User study template.** Each participant is asked to choose an image that best meets two criteria: subject fidelity and text prompt fidelity. We asked 50 participants to answer 30 questions each on Amazon Mechanical Turk and show the win-rate.

To evaluate real-world user preferences for various personalization methods, we conducted a user study using Amazon Mechanical Turk. We recruited 50 participants, each of whom answered 30 questions. Participants were instructed to select the image that best satisfied two criteria simultaneously: (1) resembling the given subject and (2) adhering to the provided prompt. For each question, users compared two images: one generated by our method and the other by a prior work. The prior methods used for comparison included Custom Diffusion (Kumari et al., 2023), NeTI (Alaluf et al., 2023), and DisenBooth (Chen et al., 2024b), which represent the most recent advances in this domain. Example questions are provided in Fig. 12.

To ensure a fair comparison, output images were randomly chosen (random seed and prompt) from the DreamBooth Ruiz et al. (2023) prompt set. When comparing two images (as shown in Fig. 5 options (a) and (b)) we used the same random seed. Furthermore, to avoid position bias, the image options were presented to participants in a randomly shuffled order. The win rates for each method were calculated and are summarized in Fig. 5 of the main paper.

## B    Comparison with Existing Works

### B.1    Positioning of our work among text-side adaptation methods

Recent work has explored the text-side of diffusion models in several different ways. Among methods that *fine-tune the text encoder*, TextCraftor (Li et al., 2024a) and TexForce (Chen et al., 2024a) show that adapting the text encoder can improve general image quality and text–image alignment, but they are not designed for instance-specific one-shot personalization. In contrast, PaRa (Chen et al., 2025) and DEFT (Kumar et al., 2025) are efficient personalization/adaptation frameworks that constrain or decompose model updates, but they are not specifically text-encoder-centered and do not address preserving the causal semantics of the text encoder. Meanwhile, a line of work operates on the *text conditioning*: P2L (Chung et al., 2023) optimizes text embeddings for inverse problems, while DATE (Na et al., 2025) and MinorityPrompt (Um & Ye, 2025) dynamically adapt text embeddings or prompts during sampling to improve minority-sample generation.

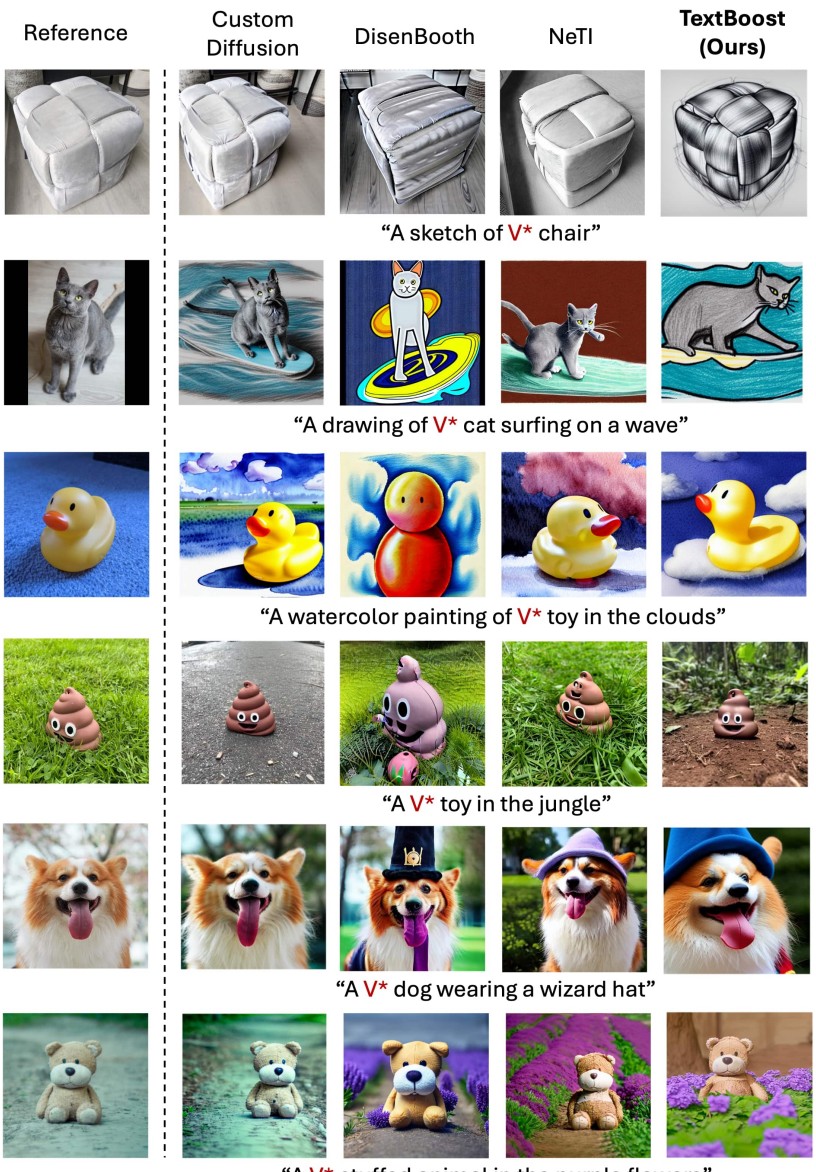

Figure 13: **More qualitative results.** We provide additional qualitative examples in comparison with recent personalization works. We used Stable Diffusion v1.5 as the baseline T2I model.

Compared with these works, our method targets **one-shot personalization** from a single reference image by selectively fine-tuning the text encoder, introducing Causality-Preserved Adaptation (CPA) to preserve the embeddings of tokens before V⋆, and further enhancing conditioning with layer-wise embeddings injected before each cross-attention layer.

## B.2    More experiments for comparison

Here, we present more qualitative results compared to several recent works, as illustrated in Fig. 13.

For our baseline text-to-image (T2I) model, we employed Stable Diffusion v1.5. It is important to note that the results from Stable Diffusion v2.1-base and v2.1 are discussed in the main paper. Our method

demonstrates a superior ability to generate images that closely align with the subjects of the reference images, significantly outperforming other approaches in capturing the essence of the given text prompts.

In the context of style personalization (i.e., stylization), we compared our approach with DisenBooth, as detailed in the main paper. Additionally, we explored stylization using NeTI (Alaluf et al., 2023), though these results were not included in the main paper due to NeTI's tendency to rapidly overfit. This overfitting results in the memorization of style, often at the expense of properly reflecting the input text prompts. A comparison with NeTI is provided in Fig. 14.

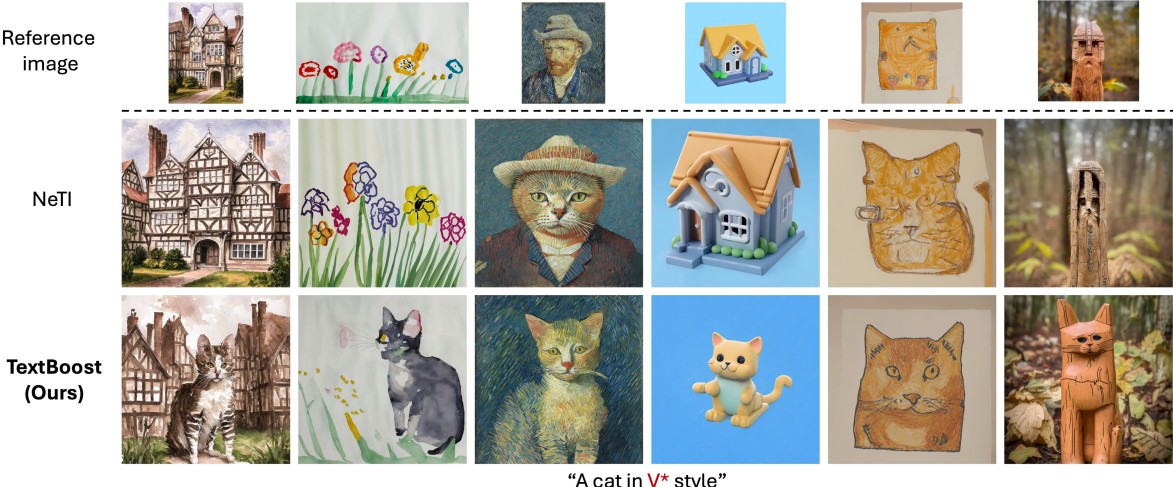

Figure 14: **Stylization.** We provide style personalization results of NeTI and compare with our TEXTBOOST.

**Implementation Details.** We fine-tuned the baseline models following the experimental settings outlined in their respective original papers. However, since our study addresses the one-shot scenario, where only a single reference image is provided as training input, we decreased the training steps to mitigate overfitting in existing methods.

For Stable Diffusion v1.5, we set the following training steps: Custom Diffusion (Kumari et al., 2023): 250 steps, NeTI (Alaluf et al., 2023): 500 steps (without bypass), and DisenBooth (Chen et al., 2024b): 1,000 steps.

For style personalization, we used Stable Diffusion v2.1. We adjusted the batch size and training steps only if explicitly specified in the original paper; otherwise, we left them unchanged.

## C   Additional Results of TextBoost

### C.1   More results

We present additional qualitative results of our method in Figures 19, 20, 21, and 22. To demonstrate its ability to produce diverse outputs, we include four generated images for each text prompt. As shown in the figures, TEXTBOOST generates diverse images from a single reference image for a given prompt. Furthermore, it consistently produces high-quality results, offering creative control through imaginative prompts with high diversity.

Furthermore, we present an ablation study (Tab. 3) on fine-tuning the SDXL text encoders. Although fine-tuning both encoders achieves strong performance, we opt to fine-tune only the larger OpenCLIP encoder, which offers a good balance between performance and computational efficiency.

Table 3: **Effect of fine-tuning different text encoders in SDXL.** We compare personalization performance when fine-tuning only the CLIP ViT-L encoder, only the OpenCLIP encoder, or both encoders. Fine-tuning both encoders improves subject fidelity (DINO), while fine-tuning only CLIP preserves prompt adherence (VQA) better.

| Fine-tuned encoder | DINO ↑ | VQA ↑ |
|---|---|---|
| CLIP ViT-L only | 0.515 | 0.839 |
| OpenCLIP only | 0.565 | 0.763 |
| Both | 0.615 | 0.690 |

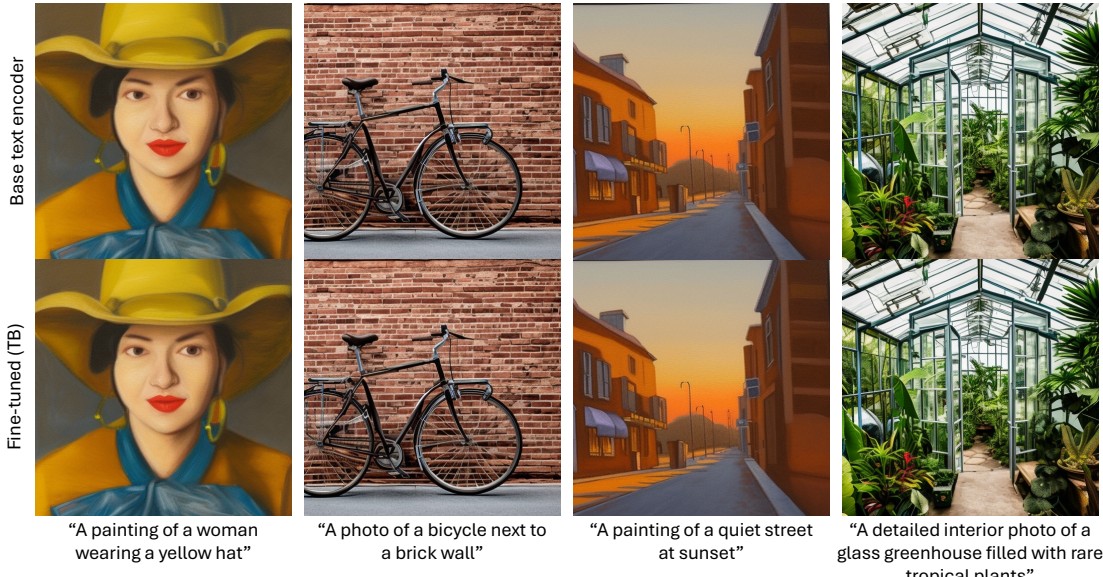

Figure 15: **Preserving text encoder behavior.** We compare images generated using the original (non-fine-tuned) text encoder and our TEXTBOOST. The results show that the original token representations are preserved, as the generated images exhibit no noticeable visual differences.

## C.2 Effect of CPA

In Fig. 3(b), we demonstrate that the proposed CPA is designed to preserve the prefix tokens that appear before $V^\star$. Although masking is intended to enforce this behavior, we further verify it empirically by measuring the cosine similarity between the prefix token representations produced by a base text encoder and those produced by TEXTBOOST (which applies CPA). The mean similarity (across all subjects) between the tokens of TEXTBOOST and the base text encoder is 1.00, indicating that the prefix representations are effectively preserved.

In addition, the behavior of the original text encoder remains unchanged for prompts that do not include $V^\star$. This is validated in the qualitative examples of Fig. 15, where the generated images remain visually identical.

Furthermore, we analyze the effect of CPA under different positions of the $V^\star$ token in the prompt. Since CPA preserves the representations of tokens preceding $V^\star$, the impact of CPA becomes more noticeable when $V^\star$ appears later in the sentence, where a longer prefix must be preserved. As illustrated in Fig. 16, CPA tends to better maintain prompt fidelity in such cases.

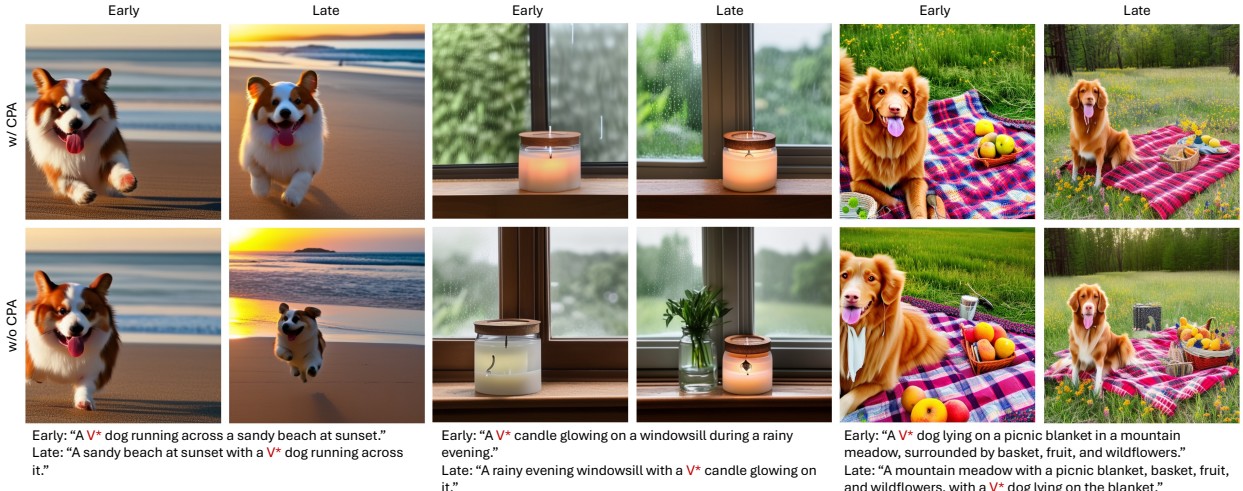

Figure 16: **Results with varying V⋆ position.** We compare generations with and without CPA when the position of V⋆ in the prompt changes. Since CPA preserves the representations of tokens preceding V⋆, its effect becomes more visible when V⋆ appears later in the sentence, where a longer prefix is preserved.

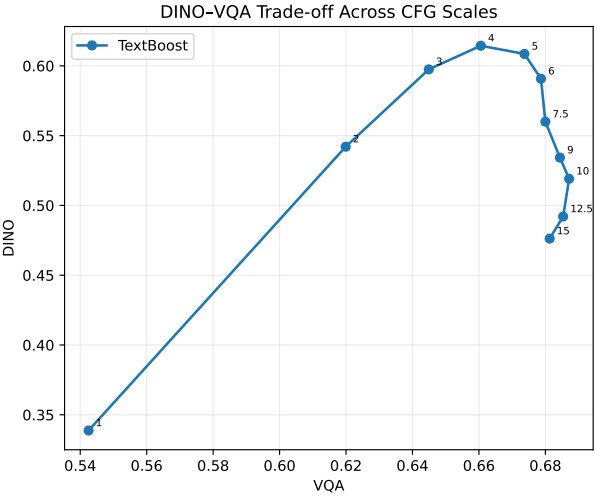

Figure 17: **DINO-VQA trade-off under a sweep of classifier-free guidance (CFG) scales.** Each point corresponds to one CFG value, so the plot shows how the operating point of TEXTBOOST changes as the conditioning strength varies.

## C.3 Trade-off across conditioning strength.

We performed a DINO-VQA trade-off experiment (Fig. 17) by sweeping the classifier-free guidance (CFG) scale at inference time. Since CFG directly controls the strength of conditioning, varying it traces an operating curve for each method rather than reporting only a single best-tuned setting. This visualization, therefore, shows how the quality of personalization changes as conditioning is strengthened or relaxed.

## D Additional Results of TextBoost++

To provide a comprehensive comparison of text encoder tuning approaches (naive tuning, TEXTBOOST, and TEXTBOOST++), we present generated images with diverse subjects and prompts in Fig. 18. As shown in the

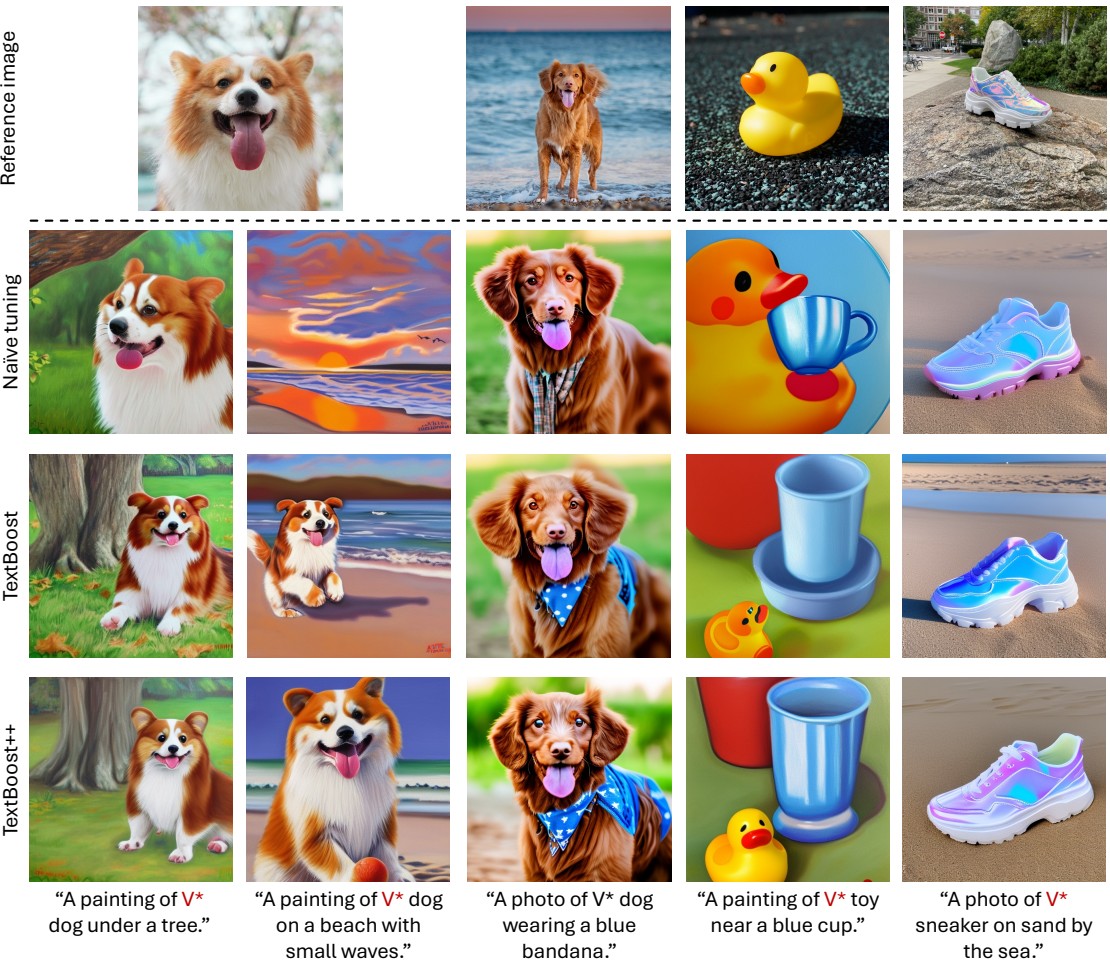

Figure 18: **Comprehensive comparison.** We compare naive text encoder fine-tuning, TEXTBOOST, and TEXTBOOST++. Across the generated images, tb improves prompt adherence compared to naive tuning, while TEXTBOOST++ improves subject fidelity.

figure, TEXTBOOST improves text prompt fidelity compared to naive tuning. Furthermore, TEXTBOOST++ achieves better subject fidelity than tb, which is consistent with the results reported in Tab. 1.

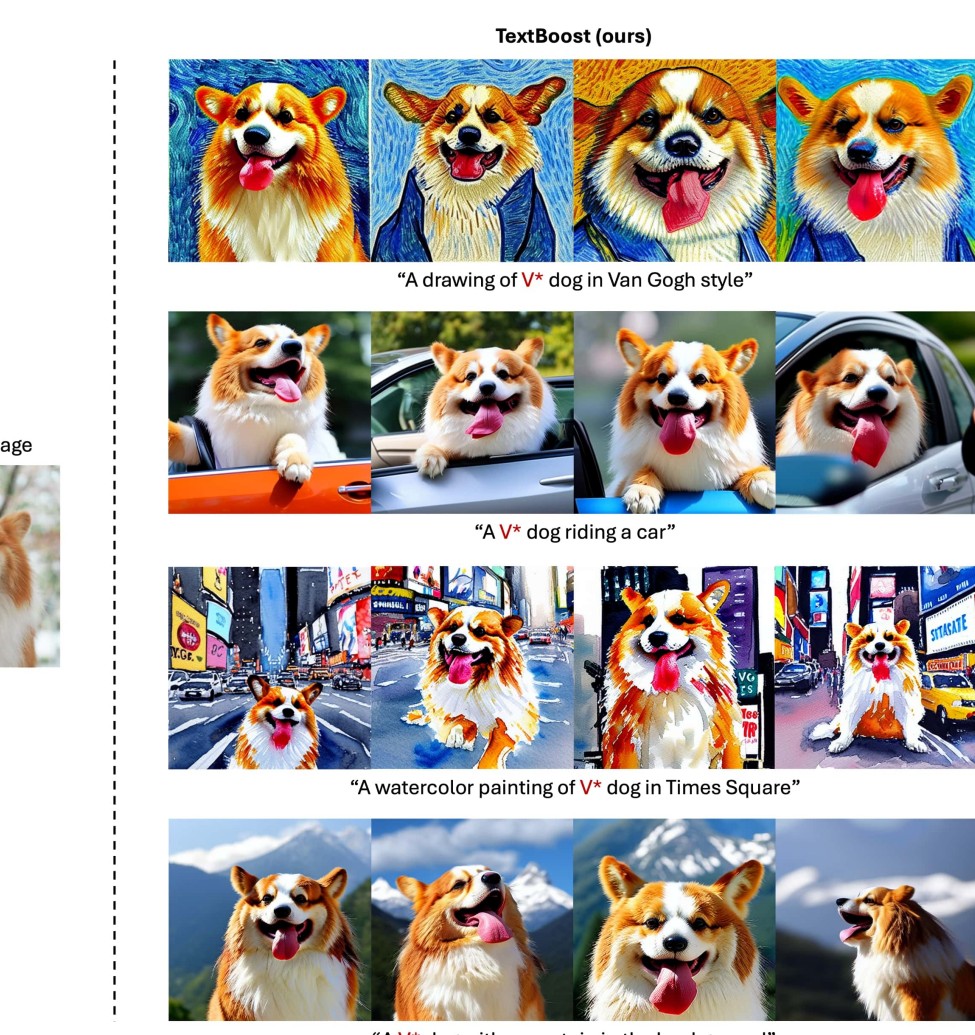

Figure 19: **More qualitative results of our TextBoost (dog).**

a single
reference image

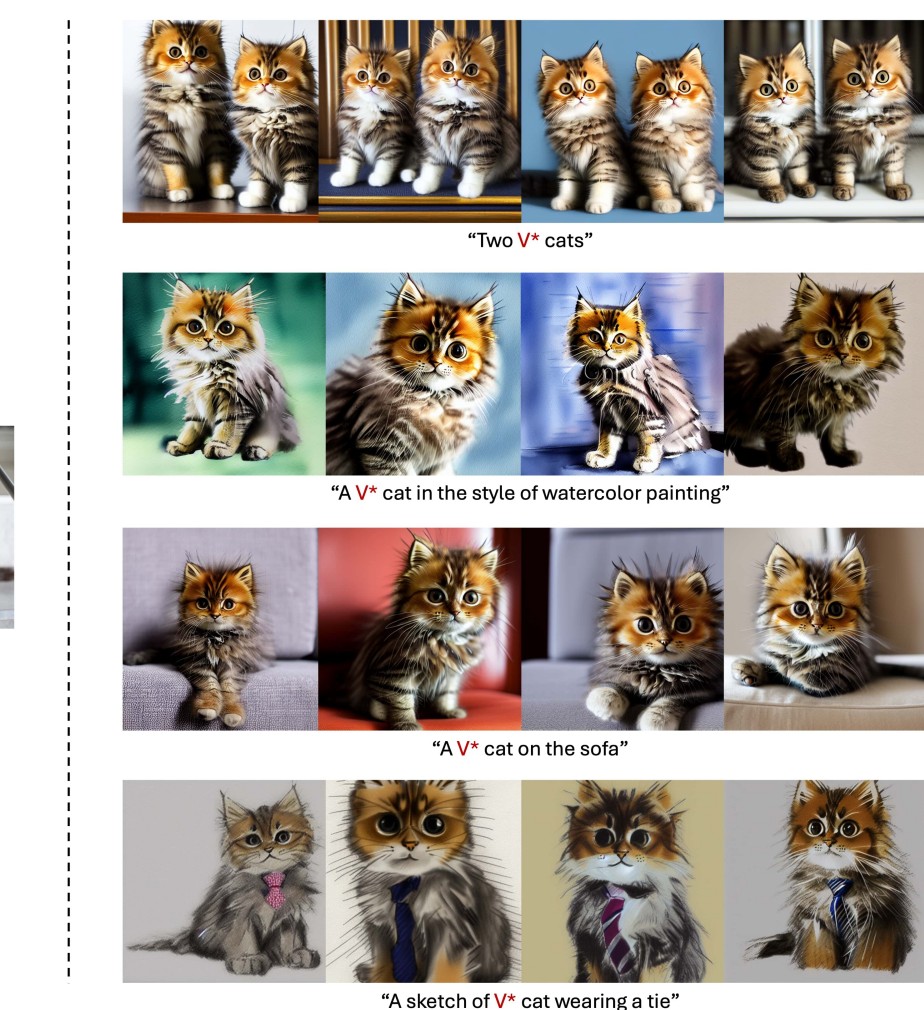

"Two V* cats"

"A V* cat in the style of watercolor painting"

"A V* cat on the sofa"

"A sketch of V* cat wearing a tie"

Figure 20: **More qualitative results of our TextBoost (cat).**

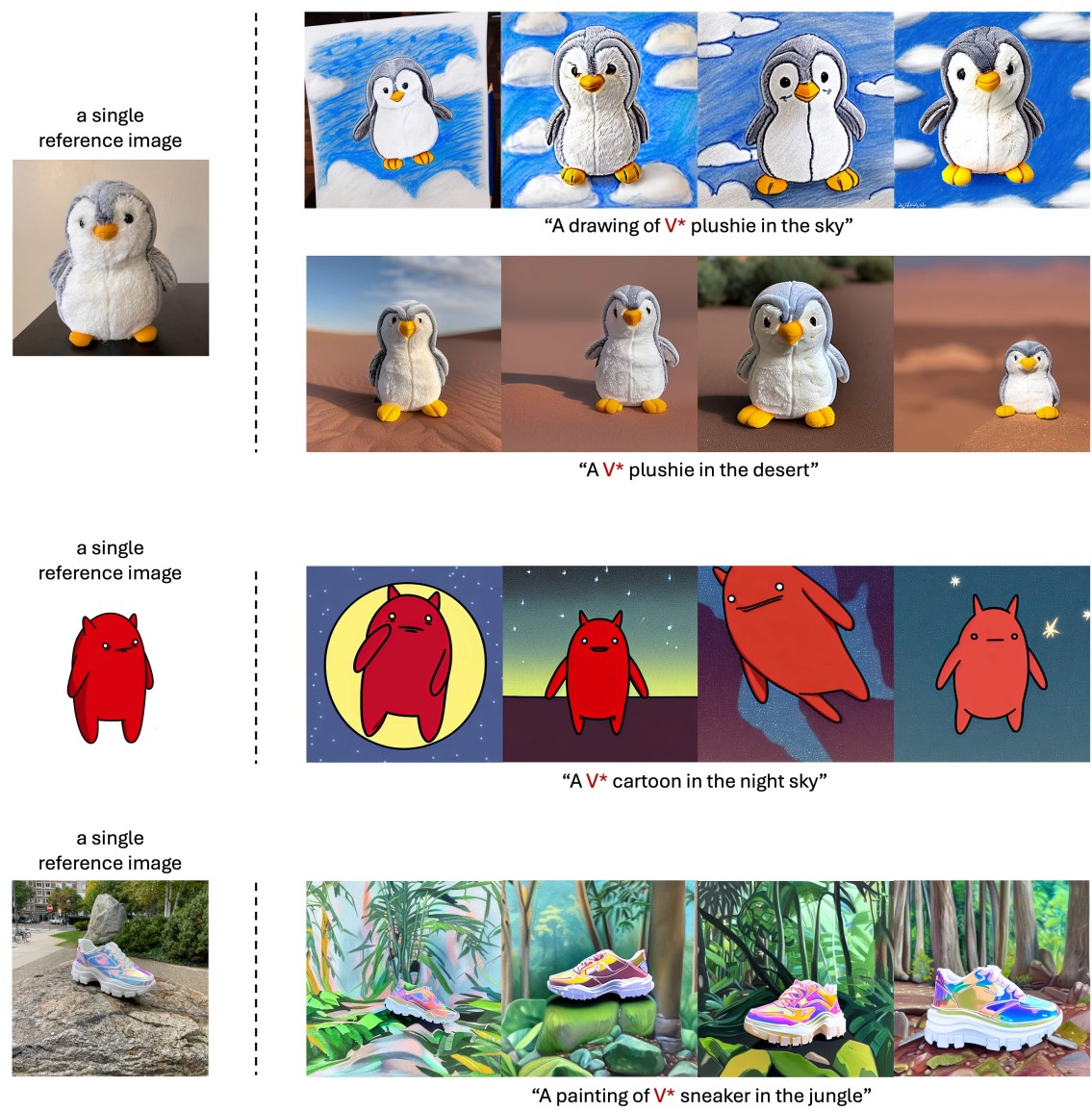

Figure 21: **More qualitative results of our TextBoost (several subjects).**

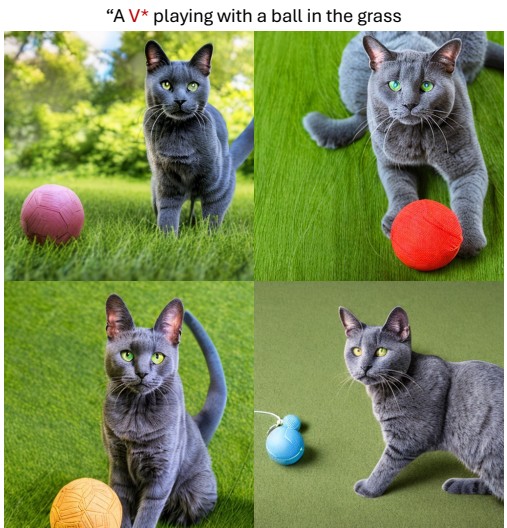

Figure 22: **Diversity results.** We show four generated outputs of our method with the same input text prompt. Our method, TEXTBOOST, consistently generates images with high diversity.

# E   Experiments on Human Faces

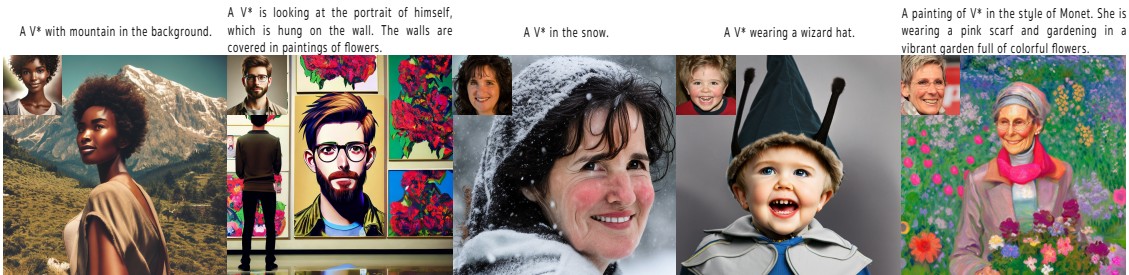

Figure 23: **Qualitative results on human faces.** We use 3 random faces from FFHQ and 2 generated images from DALL-E. We then use a single reference face image to train our TextBoost. The reference face image is located at the upper-left corner of each generated output.

To demonstrate the versatility of TextBoost on a broader range of reference images, we conduct experiments using facial images. To avoid using celebrity images, we select three random faces from FFHQ and two synthetic faces generated by DALL-E. The results, shown in Fig. 23, highlight that even with complex captions, TextBoost produces high-fidelity images with strong text-image alignment, demonstrating clear advantages of our approach.

