# OpenReview forum: "Boosting Text Encoder for Personalized Text-to-Image Generation"
_TMLR — Accepted by TMLR_

### Review · Reviewer_GrJP · 2026-02-13

**Summary Of Contributions:**

The paper introduces TextBoost, a lightweight, one-shot method for personalization of T2I diffusion models by selectively finetuning the text-encoder. It begins by pinpointing the fact that the text encoder is very sensitive to perturbations. Common wisdom so far is that since the U-net is responsible for the visual diffusion process, personalization is mainly happening by finetuning the U-net parameters. However, the authors have realized that the layers of the text encoder show significant updates when finetuning the T2I model, indicating that updates to the text encoder hold more potential. By only needing to update the adapters added to the text encoder, TextBoost only contains 0.12M-0.15M params to be trained, making it more suitable for practical real-world use cases.
The authors present 2 main contributions to the text encoder:

1. The authors take inspiration from adapter methodologies, and introduce CPA, the causality preserved adapter. The main argument made is that by masking the tokens up until the V* token, the model only updates itself from the V* token onwards. The goal of CPA is to update as little as possible in the already brittle text-encoder and only learn how the new concept token fits into the subsequent tokens.
2. Extended textual embedding: the authors take inspiration from related works in GANs and implement extended textual embeddings for the CLIP text encoder, by adapting the text embeddings towards each cross-attention layer of the diffusion model.
The authors evaluate TextBoost on VQA, DINOv2, and on a user study with 50 participants. The paper shows that TextBoost and TextBoost++ outperform others on VQA and are following Custom Diffusion on DINOv2. However, TextBoost is the only method that finetunes the text-encoder, also evidencing in the fact that it is the most lightweight finetuning strategy of table 1.


**Strengths:**

•	The paper introduces a very elegant and lightweight method for one-shot personalization of diffusion models and presents a method which is simple and relevant for real-world use cases since the method requires little storage requirements. This is the paper’s key strength.

•	Good prompt alignment: TextBoost achieves good VQA scores compared to other methods also backed by the results from the user study.

**Weaknesses:**

•	The paper would benefit from additional theoretical or empirical intuition to clarify why the method is effective or not. A good example is the performance drop from TextBoost to TextBoost++ on VQA. TextBoost++ improves subject fidelity but results in a noticeable drop in prompt alignment, and the authors don’t give any intuition on the reason.

•	Similarly, the authors do not provide much intuition on the effect of CPA and why it is so effective. Figure 10 is a helpful start; adding examples where naïve finetuning struggles and CPA succeeds would make the contrast more compelling and help readers better understand the benefit of the solution.

**Audience:**

Yes

**Audience Explanation:**

The paper systematically highlights a problem in current T2I models and shows that focusing on the text-encoder for diffusion personalization is an angle that has been underexplored. The method in the paper is elegant and achieves good performance with a simple change. The understanding and control over personalization approaches in T2I models is a very relevant topic. Therefore, I believe that it is beneficial for TMLR’s audience.

**Broader Impact Concerns:**

No concerns

**Claims And Evidence:**

No

**Claims Explanation:**

In general, the claims in the paper are strong and convincing. However, I feel that the paper could benefit from more elaborate evidence that builds trust and intuition for the user on why the method works. See my points below:

The related work section only discusses relevant work for personalized T2I models and PEFT methods but should have included related works as evidence that specifically focus on (low rank) parameter finetuning of the text-encoder, or alternatively, methods that also focus on one-shot personalization of diffusion models. Given the growing literature on both topics at the time of writing, bringing in a few additional representative works would help readers see how this paper connects to that momentum. Example papers that could have been discussed are TextCraftor (Li et al.) and TexForce (Chen et al.)

Related, table 1 only contains works up until 2024, and I would encourage the authors to compare against more recent works.
The authors present a user study with 30 questions across 50 participants on Amazon Mechanical Turk. The authors describe that random seeds were fixed to ensure fair image diffusion processes. The authors mention that question prompts were selected as "diverse subjects and text prompts sourced from Ruiz et al.". It would be helpful to briefly describe how the prompts were selected (e.g., random with a fixed seed or curated), so readers can better assess the study design. With only 30 questions, it could be that prompts were selectively chosen towards well working cases. At present, the paper doesn’t provide evidence that rules out possible prompt selection effects; a short clarification would address this concern.

Chen, C., Wang, A., Wu, H., Liao, L., Sun, W., Yan, Q., & Lin, W. (2024, September). Enhancing diffusion models with text-encoder reinforcement learning. In European Conference on Computer Vision (pp. 182-198). Cham: Springer Nature Switzerland.
Li, Y., Liu, X., Kag, A., Hu, J., Idelbayev, Y., Sagar, D., ... & Ren, J. (2024). Textcraftor: Your text encoder can be image quality controller. In Proceedings of the IEEE/CVF Conference on Computer Vision and Pattern Recognition (pp. 7985-7995).

**Requested Changes:**

While researching related works in the domain of one-shot personalization of diffusion models, I have stumbled upon the paper by Park et al. which is also called TextBoost. The figures in the papers suggest that this original paper comes from the same authors and targets the same problem, therefore I believe this is an initial implementation by the same authors. However, the methodology between that paper differs from this one and does not include the 2 main contributions of this paper. I see that the original paper already has been cited, and Seo et al. reference the SNR-based sampling approach of the original paper. Therefore, my suggestion for the authors of this paper is to change the method name, to avoid further confusion in the future.

As explained in one of the earlier questions, I think the paper would benefit from a more complete related work section with a comparison with other text-encoder finetuning methods and/or one-shot personalization methods.

The CPA method is introduced to with the goal of maintaining the auto-regressive causality or the text-encoder. Adding targeted qualitative evidence would help substantiate this claim. An experiment that would clearly show the effect of CPA would be a qualitative comparison with- and without CPA. It would be effective to see (in an image) that the model updates intended to teach the model a new subject (V∗) can ripple backward through the model's weights and interfere with the model’s semantic integrity of the preceding tokens. For instance, prompts where V∗ appears late in the sentence could make any causality effects easier to observe. So instead of: "A painting of V* toy swimming in a pond in the snowy winter, where the pond is filled with fish and the sky is filled with stars.", I would want to see: "A painting of a pond in the snowy winter, where the pond is filled with fish and the sky is filled with stars and a V* toy swimming in it." If CPA is crucial for maintaining all the details up until the concept token, it should become clearly visible with such a detailed, descriptive prompt.

The paper emphasizes on the finetuning of the text encoder over the finetuning of the U-net. While the arguments for this are strong (both the fact that it’s sensitive to change and able to be finetuned in a lightweight way), I see examples of images in the qualitative results that have lost some aspects of their fidelity. E.g. Fig. 6: “A V* dog flying in the universe”, we see that the white stripe on the dog’s head has changed into an asymmetric stripe. Also, the result in table 1 support this claim as it ranks slightly lower in the DINOv2 score compared to competitors. Therefore, it would be very interesting to see the effect of combining TextBoost with U-net finetuning methodologies (Similar to TextCraftor, who show that fine-tuning the text-encoder is orthogonal to the U-net and that finetuning both lead to superior performance). A possible experiment could be to implement TextBoost on top of other methods such as Custom Diffusion, purely to see to what extent the results can be improved (both qualitatively and quantitatively). Of course, it is evident that it would lose its lightweight edge, but it would show its robustness and generalizability. In general, it would be good for the reader to understand the generalizability of the method towards other architectures.

In the user study, a one sentence note on prompt selection (and seed, if applicable) would improve clarity and reproducibility for readers and build trust that prompts are not hand selected.

Seo, H., Jeong, W., Lee, K., & Chun, S. Y. (2025). Efficient Personalization of Quantized Diffusion Model without Backpropagation. In Proceedings of the Computer Vision and Pattern Recognition Conference (pp. 7717-7727).

---

> ### Author Response · Authors · 2026-03-09
>
> Thank you for highlighting the strengths of our work, especially the elegance and lightweight nature of the method, its practical relevance due to very low storage requirements, and its strong prompt alignment in qualitative and quantitative results.
>
> ### **Method name and related work**
> Thank you for raising the concern about possible naming ambiguity.
> Because the submission is currently under double-blind review, we cannot discuss potential external preprints or authorship inference in the public discussion.
> Separately, we expanded the related-work discussion in the revised manuscript (Appendix B.1) to clarify the position of the paper relative to recent text-side methods.
>
> ### **Why TextBoost++ improves subject fidelity but lowers VQA**
> We have added direct comparison results in Appendix D. The additional layer-wise conditioning in TextBoost++ increases subject-specific capacity, which improves subject fidelity, but can induce generation toward the personalized concept in highly compositional prompts. We therefore present TextBoost and TextBoost++ as two complementary operating points rather than one uniformly dominating the other.
>
> ### **Experiment on identifier token position**
> Following your suggestions, we have included the qualitative comparison in Fig.16 and discuss it in Appendix C.2. The results suggest that the effect of CPA becomes larger when $V*$ token is placed near the end of the prompt.
>
> ### **Combination with U-Net fine-tuning methods**
> We appreciate this suggestion and agree that the two directions are complementary. At the same time, we note that our paper’s goal is to present a lightweight standalone personalization method, which is why the paper emphasizes the text-encoder-only setting and parameter efficiency.
> As suggested, we now state more clearly that our approach is orthogonal to U-Net-side personalization, with the results shown in table below, where the combination further improves the subject fidelity.
>
> | Method | DINO | VQA |
> |---|---|---|
> | TextBoost (TB) | 0.560 | 0.680 |
> | TB + U-Net tuning | 0.608 | 0.680 |
>
> ### **User-study prompt selection**
> The prompts were randomly selected from the DreamBooth (Ruiz et al.) template pool, random seeds were fixed across methods, and the displayed options were shuffled during the study for fair comparison. In the revised manuscript, we added this detail to ensure transparency.

---

### Review · Reviewer_uGBx · 2026-02-22

**Summary Of Contributions:**

The authors propose a one-shot personalization method for text-to-image diffusion models that avoids fine-tuning large portions of the model. The approach has three key innovations: 1. adapting the CLIP text encoder for personalization, 2. introducing Causality-Preserved Adaptation (CPA) to preserve prefix-token semantics, and 3. extending conditioning with lightweight embedding-space adapters to enable finer-grained control of cross-attention.

Strengths
1. The idea of fine-tuning the text encoder for T2I personalization is novel. To the best of my knowledge, it's previously unexplored. Most existing personalization methods focus on fine-tuning the U-Net while keeping the text encoder fixed. The authors also clearly motivate this design choice by showing the weight changes in the text encoder layers are larger than those in the U-Net in Fig. 1.

2. The authors mainly validate fine-tuning the text encoder experimentally by comparing against baselines in Table 1. and demonstrating that the approach works on SDXL in Sec. 5.5, although the experiments still require stronger controlled validation and add missing ablation studies.

3. The paper evaluates the method with a fairly large human user study, not only automatic metrics (Experimental setup). In Sec. 5.1, the authors state that they conducted a user study with 50 participants on Amazon Mechanical Turk to assess practical applicability, since automatic metrics (e.g., VQA/DINOv2) may not fully reflect real user preferences.


Weaknesses
1. The paper need to clearly explain what is Extended textual embedding

**Section 4.3 is only a high-level narrative and does not provide a formal mathematical definition of the “extended textual embedding” method.**

Without a formula, readers can’t clearly understand

a. What exactly is different between TextBoost and TextBoost++ in implementation?

b. Where the adapters sit (before K/V? before projection? on the embedding itself?)?


2. The authors still need a clean, controlled test to show that fine-tuning the text encoder is better than fine-tuning the U-Net. In Table 1, the text-encoder-based approach is compared against baseline methods, but the comparison is confounded by additional components (e.g., CPA and the embedding-space adapters). **The paper should include an ablation that isolates only text-encoder fine-tuning (without CPA and the embedding-space adapters) and compares it to a matched budget U-Net baseline** (e.g., same number of trainable parameters if applicable, same training steps, same learning rates) to demonstrate the benefit of the tuning the text encoder itself.

3. The paper still needs experiments to show CPA keeps the embeddings of the prefix tokens unchanged
In Sec. 4.2.2 (Causality-preserved adapter), authors mentioned "This mask ensures that the modifications introduced by our adapter A(x) only apply to the output representations of V∗ and subsequent tokens." However, **the paper does not provide direct empirical evidence that the output embeddings of prefix tokens remain identical or nearly identical before and after applying CPA**. For example, in example prompt "A cat in V* style", the embeddings of the prefix tokens "A", "cat", "in" should be shown to be unchanged before and after adaptation.

4. Readers may also be interested in understanding the effect of fine-tuning different text encoders in SDXL (T5 vs. CLIP vs. both).

In Sec. 5.5 (Results on SDXL), the authors note that “the larger SDXL model has two text encoders: the CLIP text encoder and a T5 model. To test the scalability of our approach, we tested TextBoost on larger models. Specifically, we fine-tuned only the CLIP encoder of SDXL, leaving the T5 model unchanged.” It would be helpful to also report results for the alternative settings: **fine-tuning T5 only, and fine-tuning both T5 and CLIP** under a matched parameter and compute budget.

5. The authors need to clearly justify the use of **different numbers of training steps when comparing TextBoost and TextBoost++**.

It seems that the paper compares TextBoost and TextBoost++ in Table 1. as an ablation study to evaluate the effect of the embedding-space adapters. However, in Sec. 5.1 (Experimental setup), the authors state that "For TextBoost++, our adapter is trained using the AdamW optimizer (Loshchilov & Hutter, 2019) with a learning rate of 1e-4 for text encoder and 1e-3 for V
∗ token for we trained for 80 steps". This difference may confuse readers and makes the comparison less controlled. Please clarify the reason for using different training-step settings, or provide results under a matched training budget.

6. The authors need to provide analysis for the adapter in each ross-attention layer contribute

The authors need to answer the following questions: **1. Do we really need adapters at every cross-attention layer? 2. are some layers more important than others?**

In Sec. 4.3 (Extended textual embedding), authors state that "To enhance the encoder’s representation capacity, we propose adapting the text embeddings individually for each cross-attention layer." However, since the U-Net contains many cross-attention layers operating at different resolutions (Low/Mid/High-resolution layers), the paper currently does not clarify which layers drive the gains. Therefore **an ablation would be to run TextBoost++ with adapters placed at: a. All cross-attention layers, b. Low-res only, c. Mid-res only, d. High-res only to quantify the contribution of adapters at each resolution**.

**Audience:**

Yes

**Audience Explanation:**

This paper is likely of interest to readers working on diffusion models, personalization, and parameter-efficient fine-tuning

**Claims And Evidence:**

Yes

**Claims Explanation:**

Although I selected Yes for this question, I still expect the authors to address **Weaknesses 2–6** by adding the missing ablation studies and the missing experimental evidence. Most importantly, please address **Weakness 1: Section 4.3 is currently only a high-level narrative and does not provide a formal mathematical definition of the proposed extended textual embedding method.**

**Requested Changes:**

Expect the authors to address **Weaknesses 2–6** by adding the missing ablation studies and the missing experimental evidence. Most importantly, please address **Weakness 1: Section 4.3 is currently only a high-level narrative and does not provide a formal mathematical definition of the proposed extended textual embedding method.**

---

> ### Author Response · Authors · 2026-03-09
>
> Thank you for recognizing the key strengths of our paper, especially the novelty of revisiting text-encoder fine-tuning for personalization, the clear motivation, the validation on larger models such as SDXL, and the inclusion of a large human user study in addition to automatic metrics.
>
> ### **Formal definition of extended textual embedding**
>
> We have now made the method fully explicit in Section 4.3. The revised manuscript defines the base hidden state $H$, the per-layer adapted embeddings $H^{(\ell)}$, and how each $H^{(\ell)}$ is used before the key/value projections of the corresponding cross-attention layer. We also explicitly state that **TextBoost = CPA only**, whereas **TextBoost++ = CPA + extended textual embedding**.
>
> ### **Text encoder vs. U-Net fine-tuning**
>
> In Table 1, we compare our text-encoder tuning approach (TextBoost) with DreamBooth-LoRA, which has the closest parameter budget among the baselines. As suggested, we additionally performed a controlled comparison between naïve text-encoder tuning and naïve U-Net LoRA tuning (LoRA rank = 1, 100 training steps). The results show that text-encoder tuning yields better prompt adherence, whereas tuning the image module yields better subject fidelity. Our proposed CPA and extended textual embedding are designed to improve the balance between these two fidelity axes.
>
> | Tuning method | DINO | VQA |
> |---|---|---|
> | Text encoder tuning | 0.522 | 0.726 |
> | U-Net tuning  | 0.606 | 0.642 |
>
> ### **Direct evidence that CPA preserves prefix tokens**
>
> We thank the reviewer for this suggestion. In the revision, we added a direct token-level analysis of representation drift before and after personalization. Specifically, we compare the baseline and personalized token embeddings and report the cosine similarity for each token position. With CPA, the mean cosine similarity of the prefix tokens is 1.00, which directly supports our claim that CPA preserves prefix semantics. We also added qualitative examples for prompts without the identifier token $V*$, discussed in Appendix C.2.
>
> Importantly, this preservation is not only empirical but also follows from the CPA design. Let $k$ denote the position of  $V*$. At each transformer layer, CPA adds an adapter output only to positions $i \geq k$, while the frozen text transformer remains causally masked. Therefore, for any prefix token $i<k$, the adapter contribution is zero, and the layer output can depend only on tokens up to position ii, none of which are modified. By induction over layers, the representations of all prefix tokens remain identical to those of the baseline encoder. Moreover, when the prompt does not contain $V∗$, the CPA mask is zero for all positions, so the model reduces exactly to the original text encoder. Hence, the text representation cannot change (up to numerical precision).
>
> ### **SDXL text encoders tuning**
>
> We corrected the factual description: SDXL uses CLIP ViT-L and OpenCLIP ViT-bigG, not T5. Accordingly, in the revised discussion we refer to CLIP-only, OpenCLIP-only, and both-encoders settings for SDXL. We compared these settings, and the results are shown below.
>
> | Tuned encoder | DINO | VQA |
> |---|---|---|
> | CLIP ViT-L only | 0.515 | 0.839 |
> | OpenCLIP only | 0.565 | 0.763 |
> | Both encoders | 0.615 | 0.690 |
>
> These results suggest that fine-tuning CLIP ViT-L gives the strongest prompt fidelity but the weakest subject similarity. Tuning both encoders improves subject fidelity at higher computational cost, but reduces VQA. Fine-tuning only the larger encoder, OpenCLIP ViT-bigG, provides the best balance at moderate cost, which is why we use this setting in our main SDXL experiment. We have added this discussion in Appendix C.1.

---

> ### Author Response · Authors · 2026-03-09
>
> ### **Different training steps for TextBoost and TextBoost++**
>
> TextBoost++ converges faster, so we used 80 steps as the default setting to improve computational efficiency. The table below shows the result of running TextBoost++ for 100 steps as well. It yields slightly higher DINO similarity with nearly unchanged VQA, at the cost of additional training steps. We also added a qualitative comparison between TextBoost and TextBoost++ in Appendix D of the revised manuscript.
>
> | Method | DINO | VQA |
> |---|---|---|
> | TextBoost++ (80 steps) | 0.598 | 0.607 |
> | TextBoost++ (100 steps) | 0.603 | 0.606 |
>
> ### **Which cross-attention layers matter most in TextBoost++**
>
> As suggested, we performed an ablation on the placement of the extended-conditioning adapters. We appended the adapters to three different ranges of cross-attention layers: early (first 4 layers), middle (middle 4 layers), and late (last 4 layers). The results are shown below.
>
> | Adapter placement | DINO | VQA |
> |---|---|---|
> | Early layers | 0.596 | 0.612 |
> | Middle layers | 0.597 | 0.605 |
> | Late layers | 0.598 | 0.604 |
> | All layers | 0.598 | 0.607 |
>
> These results suggest that early-layer adapters give the best prompt-adherence score, while adapting all layers gives the best overall balance between subject fidelity and prompt fidelity.

---

### Review · Reviewer_c2UR · 2026-02-24

**Summary Of Contributions:**

1. Text-encoder-only personalization (TextBoost): Instead of adapting the diffusion U-Net, the method personalizes by fine-tuning the CLIP text encoder, motivated by an analysis showing the text encoder tends to undergo larger effective updates during full personalization, thus you can get strong one-shot personalization with far fewer trainable parameters.

2. Causality-Preserved Adaptation (CPA): Since CLIP’s text transformer is causally masked, naïve tuning can inadvertently change embeddings for tokens before the learned concept token in a prompt. CPA introduces a masked parallel adapter so only the concept token (and later tokens) are affected, preserving prefix semantics while still learning the new concept.

3. Efficient adapter formulation: The adapter is injected in parallel to selected linear layers and combined via a token-level mask, which avoids the computationally heavier alternative of two forward passes, and keeps the personalization module lightweight and modular.

4. TextBoost++ extended conditioning: Extended conditioning by producing layer-wise refined text embeddings right before cross-attention, aiming for better fine-grained control without repeatedly re-running the full text encoder as in some hierarchical prompt-embedding approaches.

5. Experimental evaluation: Evaluated in a single reference image regime on DreamBooth-style subjects and StyleDrop-style stylization, reporting VQA-based text fidelity, DINOv2-based subject similarity and diversity (with foreground extraction), plus a MTurk preference user study comparing against multiple personalization baselines.

**Audience:**

Yes

**Audience Explanation:**

The authors build on the insights of other SoTA personalization methods and propose a more parameter efficient method - Textboost, that enables high-quality T2I personalization suitable for deployment in resource constrained real-world applications.
Researchers interested in one / few - shot image concept based T2I diffusion model personalization, and broadly image-generative models’ fine-tuning, could derive value from the findings of this paper.

**Claims And Evidence:**

Yes

**Claims Explanation:**

- Fig. 1 supports the motivation that the text encoder is a strong leverage point for personalization: during full fine-tuning, its parameters exhibit larger effective changes than the denoising U-Net, justifying a text-encoder centric adaptation strategy.

- Fig. 2 supports the need for preserving causal semantics in the CLIP text encoder: perturbations can degrade generations, motivating the design goal that adapting the learned concept token V* should not alter the output embeddings of tokens $x_i$ that precede V* in the prompt.

- Tables 1 and 2 provide quantitative evidence for both efficiency (small number of trainable parameters vs U-Net tuning baselines) and performance (text/subject fidelity and diversity metrics), supporting the paper’s empirical claim.

- Figs. 6–9 and appendix figures qualitatively reinforce the main outcomes: better subject fidelity and prompt adherence in more compositional/long prompts and stylization cases compared to recent personalization baselines.

- Fig. 10 provides ablations that connect improvements to the proposed components, including the causality-preserved adaptation (CPA) and design choices around where/how adapters are applied for CLIP text embeddings.

- The MTurk user study adds an external validation layer by comparing human preferences (win-rate) against baselines on combined criteria (subject similarity + prompt match), strengthening the empirical case beyond automated metrics.

**Requested Changes:**

- I encourage the authors to add trade-off curves in the experiments: (i) personalization/subject fidelity (e.g., DINO-based similarity) versus a divergence measure (e.g., KL between the reward-tilted posterior and the diffusion prior, or text-prompt adherence) across hyperparameters controlling conditioning strength; and (ii) performance/quality of personalization versus compute (e.g., number of trainable parameters, VRAM/memory footprint). These plots would substantiate the paper’s efficiency/performance claims by enabling Pareto-optimal comparisons of TextBoost/TextBoost++ against baselines over a sweep of settings, not only at best-tuned points.

- A more explicit mathematical formulation of the “textual embedding extension” in TextBoost++ would improve clarity. In particular, defining the layer-wise embedding construction and how it interfaces with cross-attention would help readers understand the mechanism beyond the high-level description.

- Fig. 10(b) would benefit from clarifying what each point corresponds to (e.g., which adapter placement/configuration or hyperparameter setting). Additionally, the observed trend of the VQA decreasing as DINOv2 similarity increases should be discussed, as it suggests a prompt-subject adherence trade-off that is important for interpreting the method’s behavior.

- The conceptual positioning of the proposed work would be stronger with discussion of other recent SoTA text-embedding–based personalization approaches (e.g., DEFT[1], Para[2], DATE[3], P2L[4], MinorityPrompt[5]). An empirical comparison with tradeoff curves to at least one highly relevant method (ideally DEFT given its relevance to TextBoost) would further clarify how TextBoost relates to and differs from prior text-encoder/embedding-centric SoTA.

**References**

[1] Kumar, Komal, et al. "DEFT: Decompositional Efficient Fine-Tuning for Text-to-Image Models." arXiv preprint arXiv:2509.22793 (2025).

[2] Chen, Shangyu, et al. "PaRa: Personalizing Text-to-Image Diffusion via Parameter Rank Reduction." The Thirteenth International Conference on Learning Representations.

[3] Na, Byeonghu, et al. "Diffusion Adaptive Text Embedding for Text-to-Image Diffusion Models." arXiv preprint arXiv:2510.23974 (2025).

[4] Hyungjin Chung, Jong Chul Ye, Peyman Milanfar, and Mauricio Delbracio. Prompt-tuning latent diffusion models for inverse problems. In Ruslan Salakhutdinov, Zico Kolter, Katherine Heller, Adrian Weller, Nuria Oliver, Jonathan Scarlett, and Felix Berkenkamp, editors, Proceedings of the 41st International Conference on Machine Learning, volume 235 of Proceedings of Machine Learning Research, pages 8941–8967. PMLR, 21–27 Jul 2024.

[5] Soobin Um and Jong Chul Ye. Minority-focused text-to-image generation via prompt optimization. In Proceedings of the Computer Vision and Pattern Recognition Conference, pages 20926–20936, 2025

---

> ### Author Response · Authors · 2026-03-09
>
> Thank you for the careful and accurate summary of our paper, and for highlighting the strengths of the text-encoder-only personalization idea, CPA design, the extended conditioning in TextBoost++, experimental evaluation, and practical benefit.
>
> ### **Trade-off curve**
> Thank you for this helpful suggestion. We agree that trade-off curves provide a more complete picture than reporting only a single best-tuned operating point. In the revision, we added a DINO–VQA trade-off plot in the appendix by sweeping the classifier-free guidance (CFG) scale at inference time. In this plot, DINO similarity measures subject fidelity, while VQA measures text-prompt adherence. Since CFG directly controls conditioning strength, varying it traces the operating curve of each method and makes the personalization prompt-alignment trade-off explicit. We discuss this analysis in Appendix C.2 (Fig.17).
>
> As suggested, we also performed experiments that show personalization quality versus efficiency. We note that our goal is not to claim universal dominance in every setting, but to show a favorable trade-off.
>
> | Method | Rank | DINO | VQA |
> |---|---|---|---|
> | TextBoost | 1 | 0.560 | 0.680 |
> | TextBoost | 2 | 0.594 | 0.688 |
> | TextBoost | 4 | 0.582 | 0.699 |
> | TextBoost++ | 1 | 0.598 | 0.607 |
> | TextBoost++ | 2 | 0.601 | 0.676 |
> | TextBoost++ | 4 | 0.584 | 0.696 |
>
> ### **Mathematical formulation of extended textual embedding**
> We have revised the manuscript with more explicit formulation in Section 4.3. Specifically, we define the base text embedding, the layer-wise adapted embeddings, and their injection before the key/value projections of cross-attention layer via Eqs. (7)–(9).
>
> ### **Clarification of Fig. 10(b) and VQA–DINO trend**
> Fig. 10(b) uses the same hyperparameter setting as TextBoost described in Section 5.1. Each point corresponds to a checkpoint at **20, 40, 60, 80, and 100 training steps** (ordered from right to left).
> This shows a consistent training-time trade-off: early in training, prompt adherence is stronger while subject fidelity is still limited; as training proceeds, subject fidelity improves while VQA gradually decreases. We explain that this trend is a natural subject-prompt trade-off as training proceeds.
>
> ### **Positioning relative to recent methods**
> We expanded the related-work discussion to include recent text-side methods in Appendix B.1. At the same time, we would like to note that our paper is not limited to generic text-encoder tuning alone, but proposes a causality-preserving personalization mechanism together with layer-wise extended conditioning for efficient one-shot personalization.

---

### Review · Reviewer_ohby · 2026-02-24

**Summary Of Contributions:**

The authors propose TextBoost, a one-shot personalization method that focuses on fine-tuning text encoder representations instead of the diffusion model. Over their analyses and experiments, the authors argue that in the setup of a joint optimization with diffusion model and text encoder as the trainable parameters for the personalization task, the parameters of text encoder change significantly and furthermore are crucial representations to achieve accurate personalization. Overall, the authors propose a method with respect to the causality constraints of the text encoder and with careful selection of the tuned parameters, so that the encoding performance of the text encoder is not degregaded. The authors prove the effectiveness of their approach both in terms of parameter efficiency and the various concepts provided in the dreambooth dataset.

**Audience:**

Yes

**Audience Explanation:**

This paper would serve as an interesting study for how can a text encoder such as CLIP, trained with vision alignment can serve as an effective visual representation for the personalization task. In addition, the proposed method serves as a parameter efficient alternative to the available approaches such as DreamBooth, and thus can be useful for resource constrained environments as the authors also mention.

**Broader Impact Concerns:**

The experiments are conducted on public datasets which are open for research use. In addition, for human subjects, the authors preferred samples from public datasets and generated datasets. Furthermore, such an impact statement is not necessary.

**Claims And Evidence:**

Yes

**Claims Explanation:**

- The experiments are presented with a standardized personalization benchmark, Dreambooth dataset, with sensible choices for the evaluation metrics.
- Since the method is suited as a fine-tuning mechanism for a personalized concept, the available comparisons are fair, which involve finetuning the model in different ways for this task.
- The presented experiments measure the text-image alignment, and concept accuracy in a cinvincing way.
- The authors include a user study as a perceptual metric.
- However, as mentioned in the requested changes, the cali on perserving the text encoding structure needs additional experiments to be backed up.

**Requested Changes:**

- In the section for the results with SDXL, the authors claim that SDXL uses CLIP and T5 as the text encoder. This is factually wrong, where SDXL uses OpenCLIP and CLIP as the two text encoders. In addition, the setup for FLUX includes the mentioned text encoders, instead of SDXL. With that mention, if it is claimed that the method works for encoders like T5, the authors should provide examples with FLUX, to back up the claim. This comparison would
- The paper mentions the overall method as TextBoost, where they manipulate the text encoder for personalizaed image generation. However, the final approach is the mentioned as TextBoost++, which uses the extended embedding space. The authors are encouraged to explain how this extended space should be utilized in comparison to the vanilla version. As an example, are the different examples given to different layers of the cross attention, or concatenated and used as a single joint representation.
- Regarding the adapter placement, the authors provide ablations on the moditied components of the transformer blocks in Fig. 10. In addition to the analysis provided here, I believe experiments on the blocks modified would back up the architectural choices for including the adapters.
- The authors claim that their personalization method is motivated by preserving the embedding structure of the baseline text encoder. However, the provided experiments are more focused on the visual quality with the personalizaed embeddings. The authors should consider showing how does the performance on text encoding change (e.g., does the representation quality change if you don't include the identifier token V*).
- WHile this is not a compulsory request given the scope of the paper, the experiments section would benefit from comparisons with human centered samples (e.g. human faces), where the concept specific details are high.

---

> ### Author Response · Authors · 2026-03-09
>
> Thank you for recognizing the strengths of our paper, especially the strong empirical evaluation, the fairness of the comparisons, the inclusion of a user study, and the practical value of our parameter-efficient approach for resource-constrained settings.
>
> ### **SDXL / FLUX clarification**
>
> Thank you for catching this. We corrected the manuscript to state that SDXL uses CLIP ViT-L and OpenCLIP ViT-bigG, and in our SDXL experiments we fine-tuned only the OpenCLIP encoder while keeping CLIP ViT-L frozen. We also removed wording that could be read as a claim about T5-based validation, since the current paper does not include FLUX experiments.
>
> ### **Clarification of TextBoost vs. TextBoost++**
> We now make this distinction more explicit in Section 4.3. **TextBoost** denotes CPA-based text-encoder personalization, while **TextBoost++** augments it with layer-wise extended textual embeddings. Specifically, each cross-attention layer receives its own adapted embedding before the K/V projections, rather than using a single concatenated representation. We also added the explicit mathematical formulation (Section 4.3) and qualitative comparison (Appendix D) in the revised manuscript.
>
> ### **Adapter placement / architectural choice**
> In addition to the original fc2 placement study in Fig. 10(b). As suggested, we performed ablation on the modified blocks for the extended-conditioning module. We appended the adapter in three ranges (early: first 4 layers, middle: middle 4 layers, end: last 4 layers).
>
> | Adapter placement | DINO | VQA |
> |---|---|---|
> | Early layers | 0.596 | 0.612 |
> | Middle layers | 0.597 | 0.605 |
> | Late layers | 0.598 | 0.604 |
> | All layers (paper setting) | 0.598 | 0.607 |
>
> These results suggest that early-layer adapters give the best prompt-adherence score, while adapting all layers gives the best overall balance of two metrics.

---

> ### Author Response · Authors · 2026-03-09
>
> ### **Preserving the text encoding structure**
> We thank the reviewer for this suggestion. In the revision, we added a direct token-level analysis of representation drift before and after personalization. Specifically, we compare the baseline and personalized token embeddings and report the cosine similarity for each token position. With CPA, the mean cosine similarity of the prefix tokens is 1.00, which directly supports our claim that CPA preserves prefix semantics. We also added qualitative examples for prompts without the identifier token $V*$, discussed in Appendix C.2.
>
> Importantly, this preservation is not only empirical but also follows from the CPA design. Let $k$ denote the position of  $V*$. At each transformer layer, CPA adds an adapter output only to positions $i \geq k$, while the frozen text transformer remains causally masked. Therefore, for any prefix token $i<k$, the adapter contribution is zero, and the layer output can depend only on tokens up to position ii, none of which are modified. By induction over layers, the representations of all prefix tokens remain identical to those of the baseline encoder. Moreover, when the prompt does not contain $V∗$, the CPA mask is zero for all positions, so the model reduces exactly to the original text encoder. Hence, the text representation cannot change (up to numerical precision).
>
> ###  **Face experiment**
> Face-personalization results can be found in Appendix E, using faces sampled from FFHQ and synthetic faces (DALLE2). These examples show that our method also preserves fine-grained facial details in a challenging personalization setting (simple to complex prompts).

---

### Author Response · Authors · 2026-03-09
**General response**

We thank all reviewers for their positive assessment of our work and for their valuable feedback. We have revised the manuscript to address each of the reviewers’ suggestions and comments. All modifications in the revised paper are **highlighted in blue**. When an entire subsection has been newly added, the corresponding subsection title is also marked in blue.

Once again, we sincerely appreciate the reviewers’ time and thoughtful comments, which have helped improve the clarity and quality of our paper.

---

### Decision · Action_Editor_ote9 · 2026-04-01

**Recommendation:** Accept with minor revision

**Additional Comments:**

The authors should make the distinction with the Arxiv version clear in the camera ready version. Also, the draft will benefit from a final proof-read before publication.

**Audience:**

Yes

**Audience Explanation:**

As described earlier, the topic is both timely and relevant and the results are interesting. Overall, a worthwhile addition to the literature and useful for the community.

**Claims And Evidence:**

Yes

**Claims Explanation:**

From reviewers' perspective, the paper is consistently viewed as addressing an important and underexplored aspect of personalization in text-to-image diffusion models, namely, the role of the text encoder and conditional embeddings. Several reviewers highlight the novelty and practicality of lightweight text-encoder adaptation, solid empirical validation, and improved clarity after revision (including mathematical formulation, trade-off analyses, and strong experimental evidence). Most major concerns were resolved, particularly around formulation, efficiency, and validation of claims such as preserving prefix semantics and outperforming U-Net fine-tuning. However, reviewers also point out remaining weaknesses: limited evaluation on high-fidelity domains (e.g., faces), reliance on relatively outdated or few encoder variants (e.g., limited CLIP diversity), and lack of comprehensive ablations (e.g., controlled comparisons with U-Net tuning, token-level analyses, adapter placement, and training steps). In short, with overwhelming consensus among the authors on acceptance my final decision is ACCEPT.